# Direct estimates of biomass burning $NO_x$ emissions and lifetimes using daily observations from TROPOMI

Xiaomeng Jin[1], Qindan Zhu[2], Ronald C. Cohen[1,2]

[1]Department of Chemistry, University of California, Berkeley, Berkeley, CA, 94720, United States
[2]Department of Earth and Planetary Science, University of California, Berkeley, Berkeley, CA, 94720, United States

*Correspondence to*: Xiaomeng Jin (xiaomeng_jin@berkeley.edu) and Ronald C. Cohen (rccohen@berkeley.edu)

**Abstract.** Biomass burning emits an estimated 25% of global annual nitrogen oxides ($NO_x$), an important constituent that participates in the oxidative chemistry of the atmosphere. Estimates of $NO_x$ emission factors, representing the amount of $NO_x$ per mass burned, are primarily based on field or laboratory case studies, but the sporadic and transient nature of wildfires makes it challenging to verify whether these case studies represent the behavior of the global fires that occur on earth. Satellite remote sensing provides a unique view of the earth, allowing the study of emissions and downwind evolution of $NO_x$ from a large number of fires. We describe direct estimates of $NO_x$ emissions and lifetimes for fires using an exponentially modified Gaussian analysis of daily TROPOspheric Monitoring Instrument (TROPOMI) retrievals of $NO_2$ tropospheric columns. We update the *a priori* profile of $NO_2$ with a fine-resolution (0.25˚) global model simulation from NASA's GEOS Composition Forecasting System (GEOS-CF), which largely enhances $NO_2$ columns over fire plumes. We derive representative $NO_x$ emission factors for six fuel types globally by linking TROPOMI derived $NO_x$ emissions with observations of fire radiative power from Moderate Resolution Imaging Spectroradiometer (MODIS). Satellite-derived $NO_x$ emission factors are largely consistent with those derived from in-situ measurements. We observe decreasing $NO_x$ lifetime with fire emissions, which we infer is due to the increase in both $NO_x$ abundance and hydroxyl radical production. Our findings suggest promise for applying space-based observations to track the emissions and chemical evolution of reactive nitrogen from wildfires.

## 1 Introduction

Biomass burning emissions affect global radiative forcing, the hydrological cycle, ecosystem and air quality (e.g., Crutzen and Andreae, 1990; Penner et al., 1992; Johnston et al., 2012; Liu et al., 2014). Biomass burning emits an estimated 25% of global annual nitrogen oxides ($NO_x$ = NO + $NO_2$), an important constituent that participates in the oxidative chemistry of the atmosphere, leading to the formation of tropospheric ozone ($O_3$) and secondary aerosols that affect air quality, ecosystem health and climate. Unlike other $NO_x$ sources such as power plants that are persistent and relatively static, the sporadic and transient nature of wildfires makes it challenging to estimate emissions experimentally over wide spatial and temporal scales (Ichoku and Ellison, 2014). Biomass burning emissions inventories used in models are subject to uncertainties in estimates or measurements of the burned area, fuel loadings, combustion efficiency, and also the compound-specific emission factors that

relate the mass of a chemical species emitted to fuel consumption (e.g., Petrenko et al., 2012; Liu et al., 2020; Carter et al., 2020). Current estimates of $NO_x$ emission factors are primarily based on field measurements that sample a few fires over a small region (Yokelson et al., 2007; Alvarado et al., 2010; Lindaas et al., 2021), or laboratory studies that measure fire emissions under controlled conditions (McMeeking et al., 2009; Roberts et al., 2020). These previous studies report large variations of $NO_x$ emission factors, even in a similar ecosystem, which could be due to variation in individual fire conditions, nitrogen content of the fuel burned, or differences in sampling techniques and analysis methods (Andreae, 2019). The $NO_x$ emission factors used in global biomass burning emission inventories also vary (Wiedinmyer et al., 2011; Kaiser et al., 2012; Darmenov and Silva, 2015; van der Werf et al., 2017). These varying perspectives raise the question of how well we understand how to extrapolate the emission factors derived from individual fires to the large number of fires that occur annually on the globe, each with distinct fire conditions, intensity, and fuel type.

Satellite remote sensing provides a unique view of the earth, which offers the opportunity to study a large number of fires globally. Satellite-based products such as fire radiative power (FRP) and burned area have been widely used in the fire detection (Wiedinmyer et al., 2011; van der Werf et al., 2017). The launch of GOME-1 in 1995 set the milestone for monitoring $NO_x$ from space (Richter et al., 2005). The Ozone Monitoring Instrument (OMI) onboard Aura satellite has a finer spatial resolution with $13 \times 24$ $km^2$ at nadir, which is widely used to detect $NO_x$ emissions from anthropogenic sources (Beirle et al., 2011; Lu et al., 2015; Duncan et al., 2016; Liu et al., 2016). These studies typically rely on aggregation of multiple observations to reduce the noise of satellite retrievals or improve spatial resolutions via oversampling, tools which cannot be used to study the rapidly varying $NO_x$ from fires. In October 2017, the TROPOspheric Monitoring Instrument (TROPOMI) launched to space (Veefkind et al., 2012). The finer spatial resolution ($\sim 7 \times 3.5$ $km^2$), and the improved signal-to-noise ratio of TROPOMI compared to OMI offer new opportunities to more reliably interpret observations of individual plumes (Veefkind et al., 2012; Judd et al., 2019; van Geffen et al., 2020).

The accuracy of satellite retrieval of $NO_2$ columns largely depends on the *a priori* knowledge of $NO_2$ vertical profile shape needed for calculating air mass factor (e.g., Boersma et al., 2018; Verhoelst et al., 2021). The impacts of the *a priori* profile are especially evident for fire plumes with intense emissions and varying plume dynamics (Bousserez, 2014). Previous studies that use satellite observations to derive $NO_x$ EFs from fires all show lower EFs than *in situ* measurements, which could be due to inaccurate *a priori* profiles (Mebust et al., 2011; Mebust and Cohen, 2014; Schreier et al., 2015). Replacing the *a priori* vertical profile from a fine-resolution regional model can enhance the spatial gradient and correct the low bias of satellite retrieved $NO_2$ over polluted regions (e.g., Russell et al., 2011; Valin et al., 2011; Goldberg et al., 2017; Ialongo et al., 2020; Judd et al., 2020; Tack et al., 2021). However, conducting fine-resolution simulations for fires distributed globally is currently too computationally expensive for routine analysis. The GEOS Composition Forecasting System (GEOS-CF) produced by NASA Global Modeling and Assimilation Office (GMAO) provides real-time global simulations of atmospheric composition at a fine resolution of 0.25° resolution (Keller et al., 2021). The GEOS-CF has provided an opportunity for capturing fine-scale features relevant to biomass burning plumes. Here we apply GEOS-CF simulated $NO_2$ as the *a priori* profile to re-calculate AMFs for TROPOMI $NO_2$ columns near fires, and we show updating the *a priori* profile could resolve the underestimate of

satellite-based $NO_x$ emission factors suggested in previous studies (Mebust et al., 2011; Mebust and Cohen, 2014; Schreier et al., 2015).

Satellite instruments observe fire $NO_x$ plumes as a mixture of fresh and aged smoke. $NO_x$ is a short-lived species, and its concentration will decay in the plume due to the formation of nitric acid ($HNO_3$), peroxyacetyl nitrate (PAN) and organic nitrates ($RONO_2$). The relationship between satellite observed $NO_x$ concentration and emissions depends on the loss rate of

$NO_x$. The chemical processes governing the lifetime of $NO_x$ in the fire plumes are poorly understood (Alvarado et al., 2010). Previous studies assume a constant chemical $NO_x$ lifetime of 2 hours (Mebust et al., 2011; Mebust and Cohen, 2014). Laughner and Cohen (2019) provide space-based evidence of changing $NO_x$ lifetimes over U.S. cities as $NO_x$ emissions decline. As fire intensity varies by several orders of magnitude, assuming constant $NO_x$ lifetime for all fires will likely introduce errors in the derived $NO_x$ emissions (De Foy et al., 2014). The improved spatial resolution of TROPOMI allows direct measurements of

the length scale of $NO_2$ decay. By analyzing the plume evolution downwind, we derive an effective $NO_x$ lifetime. Beirle et al. (2011) first proposed an exponentially modified Gaussian (EMG) approach to directly estimate $NO_x$ emissions and lifetimes from satellite observations, which has been widely used to derive $NO_x$ emissions from anthropogenic sources (Beirle et al., 2011; Lu et al., 2015; Goldberg et al., 2019; Laughner and Cohen, 2019). Our study is the first to apply the EMG approach to simultaneously estimate $NO_x$ emissions and lifetimes from biomass burning plumes. The resulting emission estimates provide

a large ensemble with which to evaluate current emission models and also provide detailed constraints on the chemical evolution of $NO_x$. The resulting lifetimes provide insights into hydroxyl radical abundances in the plume and thus constraints on the lifetime of other chemicals emitted from fires.

## 2 Datasets

TROPOMI is a nadir-viewing hyperspectral spectrometer launched on October 13, 2017 by the European Space Agency (ESA)

for the European Union's Copernicus Sentinel 5 Precursor (S5p) satellite mission. TROPOMI provides afternoon (~ 1:30 PM local time) global observations in the UV−visible−near infrared−shortwave spectra with a fine spatial resolution of $7 \times 3.5$ $km^2$ at nadir (increased to $5.5 \times 3.5$ $km^2$ since August 2019). We obtain the daily Level-2 TROPOMI retrievals of $NO_2$ tropospheric column density data from April 2018 to June 2020 from NASA Goddard Earth Sciences (GES) Data and Information Services Center (DISC). The retrieval of the $NO_2$ tropospheric vertical column includes three steps (van Geffen

et al., 2019): (1) retrieval of the total slant column density along the optical path using differential optical absorption spectroscopy (Boersma et al., 2011).; (2) subtraction of the total slant column density from stratospheric $NO_2$ slant column based on information from a data assimilation system (Boersma et al., 2018); (3) conversion of the tropospheric slant column density to vertical column density using air mass factors (AMFs), which are obtained from radiative transfer calculations that account for the viewing geometry, cloud fraction, surface properties, and the *a priori* vertical profile of $NO_2$ (Boersma et al.,

2018). We include TROPOMI observations with the quality assurance value higher than 0.5, which filter out problematic retrievals but still keep good quality retrievals over cloud (or aerosols). In addition to $NO_2$, we obtain TROPOMI aerosol layer

height (ALH) or pressure (ALP) data, which provides height information of aerosol layer in the troposphere. Retrieval of ALH or ALP is based on the $O_2$ absorption band at near-infrared wavelengths between 759 and 770 nm (Graaf et al., 2019). Details of the aerosol layer retrieval algorithms can be found at Graaf et al., (2019) and Nanda et al. (2019).

We use the Moderate Resolution Imaging Spectroradiometer (MODIS) Active Fire products (Collection 6) to provide information on the intensity and location of fires (Giglio et al., 2016), which are available from NASA's Fire Information for Resource Management System (FIRMS). We include daytime MODIS measurements from the Aqua satellite to match with the overpass of TROPOMI. Fire detection from MODIS is performed using a contextual algorithm that measures the infrared radiation from fires (Giglio et al., 2016). Each hotspot is recorded as the center of a $\sim 1 \times 1$ km$^2$ pixel that contains one or more

fires, and the FRP is estimated via an empirical relationship using the 4µm band brightness temperatures (Kaufman et al., 1998). We group fire pixels whose distances are within 20 km as a single fire event, and the center of the fire is calculated as the mean of fire pixel locations weighted by pixel FRP. To assess the sensitivity to the choice of FRP product, we also process the daytime Suomi NPP Visible Infrared Imaging Radiometer Suite (VIIRS) observations of active fire accessed from NASA's FIRMS. VIIRS fire product uses a similar algorithm as MODIS for fire detection (Schroeder et al., 2014). To assess potential

effects of aerosol from plumes on satellite retrieval of $NO_2$, we acquire the Multi-angle Implementation of Atmospheric Correction (MAIAC) Aerosol Optical Depth (AOD) Level-2 1-km daily gridded product (MCD19A2) from NASA's Earth Observing System Data and Information System (EOSDIS). Details of the retrieval of AOD can be found at Lyapustin et al. (2012, 2018).

The fire episodes are classified based on MODIS detected fire location following the fuel classification in the Global Fire
Emission Database (GFED), which is estimated using the MODIS land cover type product and University of Maryland classification scheme (Friedl et al., 2010; van der Werf et al., 2017). We assign the fuel type to grid cells with mixed fuel types based on the dominant fuel type. We follow the definition of GFED, grouping savanna, grassland and shrubland fires as a single herbaceous fuel type. To assess if the $NO_x$ EF varies among these three herbaceous types, we use the 500-m yearly MODIS land cover product (v5) to classify the herbaceous fires based on the dominant land cover type (Friedl et al., 2010).

We use wind fields from the hourly ERA-5 reanalysis data developed by the European Center for Medium-range Weather Forecast (ECMWF), which provides meteorological variables at $0.25° \times 0.25°$ resolution with 37 pressure levels from 1000 hPa to 1 hPa from 1979 to present (Hersbach et al., 2020). We sample ERA-5 wind data closest to the center of each fire episode at the TROPOMI overpass time (~1 PM local time).

**3 Methods**

**3.1 An improved fire *a priori* for TROPOMI NO₂**

The *a priori* vertical profiles of $NO_2$ used in the standard TROPOMI products are obtained from global daily model simulations (TM5) with coarse resolution (1°) and monthly average biomass burning emissions (Williams et al., 2017). Fires are intrinsically episodic and occur over land areas that are often as small as a few kilometers. Here we re-compute the tropospheric

AMFs using the vertical $NO_2$ profiles provided by the NASA GEOS-CF simulations with 0.25˚ resolution. GEOS-CF system

combines GEOS weather analysis and forecasting system with GEOS-Chem chemistry scheme version 12.0.1 (Bey et al., 2001; Keller et al., 2014, 2021; Long et al., 2015). GEOS-CF includes detailed gas-phase and aerosol chemistry (Knowland et al., 2020; Keller et al., 2021). The near-real-time satellite-based Quick Fire Emission Database (QFED v2.5) is used to provide daily biomass burning emissions (Darmenov and Silva, 2015). In the GEOS-CF system, 65% of biomass burning emissions are distributed within the boundary layer, and the other 35% is distributed evenly between 3.5 and 5.5 km (Fischer et al., 2014).

GEOS-CF provides hourly global vertical profiles of $NO_2$ at 23 pressure levels from 1000 hPa to 10 hPa since 2018.

We sample GEOS-CF products at the time and location of all fire episodes identified. For each episode, we spatially interpolate the GEOS-CF simulated $NO_2$ profiles to match the resolution of TROPOMI products. The AMF ($AMF_{GC, clear}$) for clear sky conditions can be calculated following Eq. (1):

$$AMF_{GC,clear} = \frac{\sum_{surf}^{trop} m_l \times x_{GC,l}}{\sum_{surf}^{trop} x_{GC,l}} \tag{1}$$

where $m_l$ is scattering weight, which is a function of satellite viewing geometry, surface pressure and reflectivity etc.; $x_{GC,l}$ is the GEOS-CF sub-column for layer $l$. We acquire averaging kernels (AK) from TROPOMI Level-2 products, and we interpolate GEOS-CF vertical profiles to the 34 vertical layers that provide information on AKs. AK is equal to the ratio of the scattering weight to the tropospheric AMFs computed from the *a priori* profile ($AMF_a$) (Eskes and Boersma, 2003):

$$AK_l = \frac{m_l}{AMF_a} \tag{2}$$

Combining Eq. (1) and Eq. (2) gives Eq. (3):

$$AMF_{GC,clear} = \frac{AMF_a \times \sum_{surf}^{trop} AK_l \times x_{GC,l}}{\sum_{surf}^{trop} x_{GC,l}} \tag{3}$$

Given that $AMF_a$ is the ratio of slant columns to vertical columns, we can relate the vertical columns with GEOS-CF simulated profile ($\Omega_{GC, clear}$) to the originally retrieved vertical columns ($\Omega_a$) as Eq. (4):

$$\Omega_{GC,clear} = \Omega_a \times \frac{\sum_{surf}^{trop} x_{GC,l}}{\sum_{surf}^{trop} AK_l \times x_{GC,l}} \tag{4}$$

For partly cloudy scenes, the air mass factor can be written as a linear combination of a clear and cloudy air mass factor (Boersma et al., 2004):

$$AMF_{GC} = f_{cloud} AMF_{GC,cloud} + (1 - f_{cloud}) AMF_{GC,clear} \tag{5}$$

where $f_{cloud}$ is radiance weighed cloud fraction, and $AMF_{GC, cloud}$ is essentially the above-cloud component of Eq. (3) (Laughner et al., 2018). Therefore, we revise Eq. (4) for the partly cloudy scene as Eq. (6):

$$\Omega_{GC} = \Omega_a \times \left[ (1 - f_{cloud}) \frac{\sum_{surf}^{trop} x_{GC,l}}{\sum_{surf}^{trop} AK_l \times x_{GC,l}} + f_{cloud} \frac{\sum_{cloud}^{trop} x_{GC,l}}{\sum_{cloud}^{trop} AK_l \times x_{GC,l}} \right] \tag{6}$$

## 3.2 Estimation of emissions and lifetimes of wildfires

We apply an EMG approach to estimate $NO_x$ emissions and lifetime from each fire episode. For each fire episode, we first rotate TROPOMI $NO_2$ swath data along the wind direction in the range of 200 km around the fire centre, and map the rotated TROPOMI $NO_2$ column to a regular grid with 0.05˚ resolution by calculating area-weighted average as described in Jin et al.
(2020). Next, we integrate the TROPOMI $NO_2$ columns in the across-wind direction within ±100 km, which gives reduced one-dimensional line densities. The $NO_2$ line densities (L) are then fitted with an EMG model, which is a convolution of a gaussian shaped emission and an exponential decay function (Beirle et al., 2011; Lu et al., 2015; Laughner and Cohen, 2019) following Eq. (7):

$$L(x|a, x_0, \mu_x, \sigma_x, B) = \frac{a}{x_0} exp\left(\frac{\mu_x}{x_0} + \frac{\sigma_x^2}{2x_0^2} - \frac{x}{x_0}\right) \phi\left(\frac{x-\mu_x}{\sigma_x} - \frac{\sigma_x}{x_0}\right) + B \tag{7}$$

where $x_0$ is the e-folding distance that represents the length scale of the $NO_2$ decay; $\mu_x$ is the location of the apparent source relative to the fire centre; $\sigma_x$ represents the Gaussian smoothing length scale; $\Phi$ is a cumulative distribution function; $a$ is a scale factor that represents the observed total number of $NO_2$ molecules in the fire plumes, and $B$ represents the background $NO_2$. We use the best guesses for initial values following Laughner and Cohen (2019). The effective $NO_2$ lifetime ($\tau_{EMG}$) and the estimated $NO_x$ emissions ($E_{EMG}$) can be calculated from the fitted $x_0$ and $a$ following Eq. (8) and Eq. (9):

$$\tau_{EMG} = \frac{x_0}{w} \tag{8}$$

$$E_{EMG} = \gamma \times \frac{a}{\tau_{EMG}} \tag{9}$$

where $w$ is the wind speed, and $\gamma$ is the ratio of $NO_x$ to $NO_2$. The effective lifetime should represent chemical lifetime of $NO_x$ if the transport speed is uniform, the direction is constant and deposition is negligible (De Foy et al., 2014). Previous studies either use the averaged wind of the first several layers (Beirle et al., 2011; Lu et al., 2015) or choose a constant layer such as
900 hPa (Mebust et al., 2011), but injection height of wildfires varies significantly, especially for large fires which inject emissions into high altitudes (Val Martin et al., 2010). To account for varying injection height, we use TROPOMI ALH as an approximation of the fire injection height instead of assuming a constant layer. We vertically interpolate ERA-5 wind data to the pressure level of aerosol layer. For the fires without valid ALH (~36% of the selected fires), we use 900hPa, as the ALP level for the majority of selected fires is near 900 hPa (see Sect. 4.1). $\gamma$ is assumed to be 1.32, which is in between measured
mean $NO_x/NO_2$ ratio of 1.50 reported in Akagi et al. (2012) and 1.24 in Juncosa Clahorrano et al. (2021). We assume a constant $\gamma$ because $O_3$ and the photolysis rate of $NO_2$ varies little in the plume, and the time scale for NO and $NO_2$ to reach steady state is of order 100s (Alvarado and Prinn, 2009). Juncosa Clahorrano et al. (2021) shows the $NO_x/NO_2$ ranges from 1.15 to 1.50 near the fire centre before 3PM LST, but the median $NO_x/NO_2$ varies little from centre to plume edge. Mebust et al. (2011) suggest the uncertainty of $NO_x/NO_2$ is ~20%.

## 3.3 Idealized plume model

To understand the factors that control the $NO_x$ lifetime, we employ a one-dimensional (1-D) multi-box plume model based on the Python Editable Chemical Atmospheric Numerical Solver (PECANS; Laughner 2019; Laughner and Cohen, 2019). PECANS is a flexible idealized atmospheric chemistry modelling framework that allows for one box to three-dimensional multi-box simulations of atmospheric chemistry with idealized transport. In this study, we set the model to be 1-D with 600 km domain size and 2.5 km resolution, which is analogous to the integrated 1-D $NO_2$ line density along wind direction. The wind speed is fixed at 5 m/s, and the diffusion coefficients are also fixed at 100 $m^2$/s following Laughner and Cohen (2019). We assume a simplified set of reactions to represent a chemical condition within the $NO_x$ plume including: (1) the permanent removal of $NO_x$ through the formation of $HNO_3$ and $RONO_2$; (2) the temporary removal and releases of $NO_x$ by PAN; (3) the instantaneous steady-state relationship between NO and $NO_x$; and (4) the transport of $NO_x$ along the wind direction. The modelled VOC are lumped into two groups; the first group (hereafter RVOC) do not contribute PAN formation; the second group is modelled as an immediate PAN precursor (hereafter OVOC), specifically acetaldehyde. We include a Gaussian shaped $NO_x$ emission source (expressed in NO) at $x$=200 km with 6 km in half width. The concentrations for $O_3$, hydroxyl radical production rate P($HO_x$), VOC reactivities, and alkyl nitrate branching ratio are given as model input. The $O_3$ concentration is fixed at 65 ppbv, which is close to observed mean $O_3$ concentration near fire plumes (Alvarado and Prinn, 2009; Alvarado et al., 2014). A fixed branching ratio to form $RONO_2$ in $RO_2$ + NO reaction of 0.05 is used following Laughner and Cohen (2019). We run PECANS repeatedly with varying $NO_x$ emissions, P($HO_x$), RVOC, and OVOC. Each model run outputs the concentration of $NO_x$ and its major sinks along the wind direction, which are then used to calculate both EMG fitted and chemical $NO_x$ lifetime.

## 3.4 Selection of wildfires

Since not all fire plumes are detectable from space, and the EMG approach works best for single sources with clear plume patterns, we apply the following four criteria to select candidate fires from fire plumes identified from MODIS FRP observations (see Fig. S1 for a flowchart of the selection procedure):

First, fires should be large enough so that an apparent enhancement of $NO_2$ is observed from TROPOMI. Considering the detection limit of TROPOMI $NO_2$ and the greater uncertainty of MODIS FRP for small fires (Kaufman et al., 1998), fire episodes with MODIS FRP higher than 200 MW are selected. We then select fires where TROPOMI $NO_2$ tropospheric columns on the fire day are at least one standard deviation higher than the mean TROPOMI $NO_2$ columns 30 days before and after the fire day (excluding the nearest four days as fires may last for several days, defined as $\Omega_{NO2\_B}$). We only include fires in which TROPOMI $NO_2$ line densities peak near the fire centre, meaning that fires with monotonically increasing or decreasing line densities within the region are excluded.

Second, nearby fire plumes should not contribute to the $NO_2$ line density of the selected fires. A major challenge of applying the EMG approach is that it only applies to single source, but isolated wildfires are rare in nature. To reduce the influence of

nearby plumes, we develop an algorithm that automatically detects and filters out the surrounding plumes (Fig. S2). We first identify plume-affected grid cells defined as $\Omega_{NO2}$ higher than background $NO_2$ ($\Omega_{NO2\_B}$, Fig. S2(b)). Next, pixels are grouped to separate plumes based on their connections with surrounding pixels, assuming that plume pixels belonging to the same fire event should be connected. We then exclude the plumes that do not belong to the centre fire plumes (Fig. S2(c)). The filtered areas are filled with background $\Omega_{NO2\_B}$ (Fig. S2(d)). The ability of clustering depends on the choice of $\Omega_{NO2\_B}$: high $\Omega_{NO2\_B}$ may truncate plumes as edges are counted as background, while low $\Omega_{NO2\_B}$ may lead to background being counted as fire plumes, so that nearby plumes are connected with centre plumes. To optimize the performance, we repeat the clustering and filtering steps with different $\Omega_{NO2\_B}$, and select the $\Omega_{NO2\_B}$ that maximizes the filtered size of nearby plumes while retains the centre plume. This filtering algorithm, however, does not apply to the case where fire plumes are overlapped. Therefore, we further exclude the fire plumes where comparable or larger fires are detected (i.e., total FRP of the nearby plumes is greater than one-third of the selected fire) over the region after applying the filtering.

Third, the fire plumes should align with the wind direction. We define a rotation bias as the angle between the wind direction and the observed apparent plume direction. From previous step, we obtain an approximate region of fire plumes, whose coordinates in $x$ and $y$ directions can be fitted with a line using linear regression, where the slope of the line can be converted to degrees (Fig. S3). We only select fire episodes with rotation biases between -30˚ and 30˚.

Fourth, the fire plumes should give good fitting statistics that satisfy the following criteria: 1) $R^2 > 0.5$; 2) $\sigma_x < x_0$, meaning that emission width is smaller than the e-folding distance, which could prevent the case in which emission shape confounds with lifetime; 3) $|\mu_x| < 50$ km, meaning that the apparent source centre is not too far from the fire centre. Fires with more than 50% missing TROPOMI $NO_2$ values are excluded. The outcome of EMG function is sensitive to the initial condition of each fitting parameter. To test the sensitivity of the fitting results to initial conditions, we repeat the fitting with varying initial values for each parameter 50 times, and we exclude fires where the standard deviation of resulting emissions is more than 50% of the emissions. After excluding the fires sensitive to initial conditions, the uncertainty of the emission due to initial conditions is ~5%.

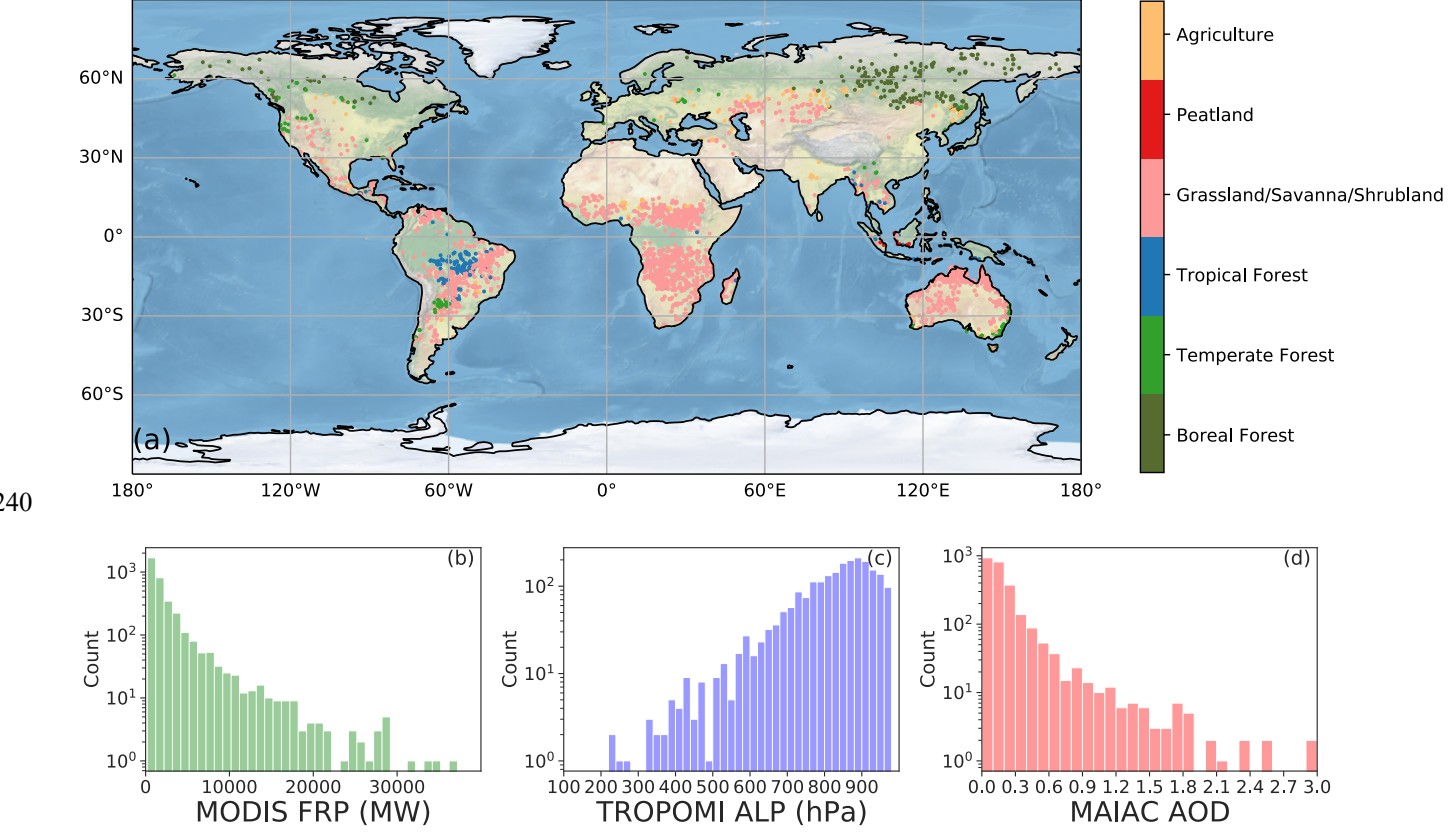

**Figure 1 (a) Map of the distribution of selected fire events and corresponding fuel type. Histograms of the distribution of (b) MODIS FRP, (c) TROPOMI ALP, (d) MAIAC AOD for the selected fire events. The number of fire events (y-axis) is on a log scale.**

## 4 Results

### 4.1 Characteristics of selected fires

Applying above selection criteria, we identified 3248 fire episodes globally between April 2018 and June 2020 suitable for the EMG approach (Fig. 1a). The majority of the fires (77%) occur in the savanna, grassland and shrubland ecosystems. We identified 573 (18%) forest fires, including 227 over the boreal forest, 153 over temperate forest and 193 over the tropical forest fires. Twenty fire episodes are classified as peatland, which occurred in equatorial Asia. We also identified 158 fires (5%) due to agriculture waste burning distributed across different regions. Figures 1b to 1d show the distributions of MODIS FRP, TROPOMI ALP and MAIAC AOD of the selected fire episodes. The MODIS FRP is below 10,000 MW for 95% of the selected fires, and 34 fires have FRP larger than 20,000 MW. The mean TROPOMI ALP is 828 hPa (or 1836 m for ALH), with $1\sigma$ standard deviation being 118 hPa (or 1751 m). Assuming TROPOMI ALH is indicative of fire injection height, more than 80% of selected fires inject fire emissions to a pressure level between 700 and 950 hPa. About 83% of the fire episodes

show MAIAC AOD less than 0.3 near the fire centre, and only 64 (3%) fire episodes have AOD higher than 1.0. In summary, most of our selected fires can be characterized as median to large fires with relatively low injection height and small AOD.

## 4.2 Emission and lifetime estimates with an example fire

We estimate emissions and lifetime for each selected fire episode. As an example, Figure 2 illustrates four major steps to estimate $NO_x$ emissions with a fire event occurred in western Australia (27.98 °S, 125.90 °E) on October 21, 2018. Several

$NO_2$ plumes are detected by TROPOMI on this day, which outperforms OMI observation on the same day which detects less smaller fires, shows less spatial gradients and larger data gap (Fig. S4). For the fire event selected, we first rotate TROPOMI observations along wind direction (Fig. 2a). Second, we update the *a priori* profile of $NO_2$ to improve the estimate of $NO_2$ column (Sect. 3.1), which leads to an enhancement of $NO_2$ gradient near the fire centre (Fig. 2b). Third, we filter two nearby fire plumes, and the nearby plumes are filled with background $NO_2$ (Fig. 2c). Fourth, we integrate across the wind direction to

obtain a 1-D line density and fit with the EMG function (Fig. 2d and Eqs. (7) to (9)). The EMG model captures the variation of the line density ($R^2 = 0.98$). The lifetime is estimated to be 1.6 hours, and the total $NO_x$ emissions are estimated as 7870 g/s.

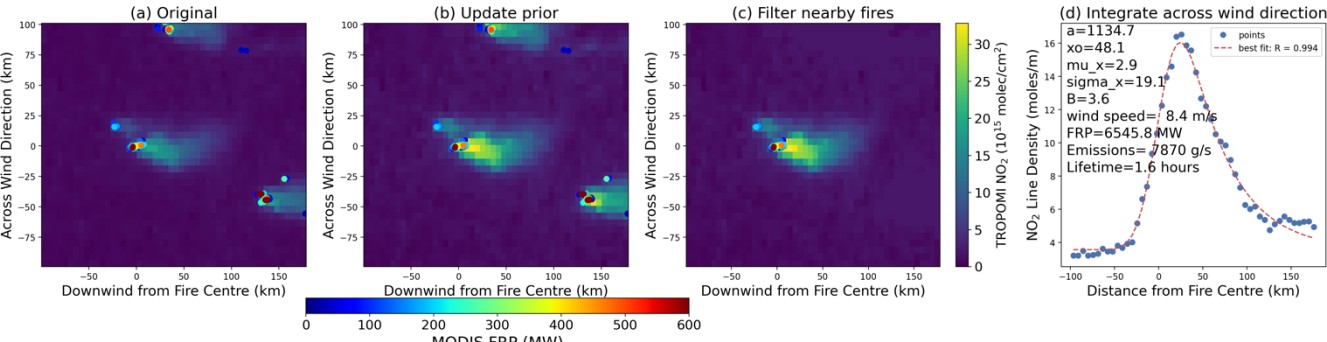

0    100    200    300    400    500    600
MODIS FRP (MW)

**Figure 2 An example fire event illustrating four steps to estimate $NO_x$ emissions and lifetime using the EMG approach: (a) original**
**TROPOMI retrieved $NO_2$ tropospheric vertical column density rotated to align with the wind direction; (b) retrieved $NO_2$ column**
**density after replacing the *a priori* vertical profile of $NO_2$ from NASA GEOS-CF simulation; (c) central fire plume after filtering the**
**nearby fire plumes; (d) $NO_2$ line density calculated by integrating $NO_2$ column density across the wind direction. The red line in (d)**
**shows the fitted line density using the EMG function (Eq. (7)). The lifetime is estimated from Eq. (8), and the total $NO_x$ emissions**
**are estimated using Eq. (9). FRP is calculated as the sum of the FPR of all fire pixels detected by MODIS shown on (c).**

## 4.3 Satellite-derived fire $NO_x$ emissions

Deriving $NO_x$ emissions for the large ensemble of fires, we investigate what drives the variation of $NO_x$ emissions and lifetimes among these fires. MODIS FRP, which represents the radiant energy released by fires, has been used to approximate the biomass burned consumption in top-down emission inventories such as Global Fire Assimilation System (GFAS; Kaiser et al., 2012). We define the emission coefficient (EC) as the mass of pollutant emitted per unit of radiative energy (i.e., Emissions/FRP), which has been used to derive the emissions of chemical species from fires (Ichoku and Kaufman, 2005;

Mebust et al., 2011; Mebust and Cohen, 2014). Figure 3 shows the relationship between TROPOMI derived $NO_x$ emissions with MODIS FRP for six different fuel types. Overall, we find TROPOMI derived fire $NO_x$ emissions are positively correlated

with MODIS FRP. We assess an overall EC by fitting a line with intercept fixed at zero. FRP explains 39% to 78% variance in emissions with the highest $R^2$ for tropical forest fires and lowest for agricultural fires. The variability not accounted for may be related to the uncertainty of satellite retrieval of $NO_2$, errors with the EMG approach, uncertainties with FRP (see Sect. 5),

and/or true differences in $NO_x$ ECs for different fires in similar ecosystems. We compute the uncertainty of EC as 95% confidence interval (CI) of the fitted EC based on the Student's $t$-distribution test. Comparing different forest types, EC is largest for tropical forests (1.30 [1.20, 1.40] g/MJ), followed by boreal forests (0.70 [0.60, 0.80] g/MJ) and temperate forests (0.56 [0.47, 0.65] g/MJ). Aggregating all grassland, savanna, and shrubland fires as a single fuel type, we obtain an overall EC of 1.02 [0.98, 1.06] g/MJ. Separating the fires to individual fuel types based on MODIS land cover classification lead to slightly

improved $R^2$ for savanna and grassland, but the derived EC is similar: 1.00 [0.93, 1.07] g/MJ for grassland, 1.13[1.06, 1.19] g/MJ for savanna and 0.89 [0.78, 1.01] g/MJ for shrubland (Fig. S5). Only 20 fires are classified as peatland fires, and we find a relatively good correlation between $NO_x$ emissions and FRP with $R^2$=0.62 and EC = 0.75 [0.47, 1.03] g/MJ. For agricultural fires, there is a large scatter between $NO_x$ emissions and FRP with $R^2$ = 0.39, and the estimated EC is 1.10 [0.88, 1.31] g/MJ. Updating the *a priori* profile of $NO_2$ enhances the spatial gradient of $NO_2$ (Fig. 2b), allowing for better estimates of fire $NO_x$

emissions. Indeed, we find that using TROPOMI standard products gives a weaker correlation between FRP and $NO_x$ emissions, and the ECs decrease by 46% on average (Fig. S6).

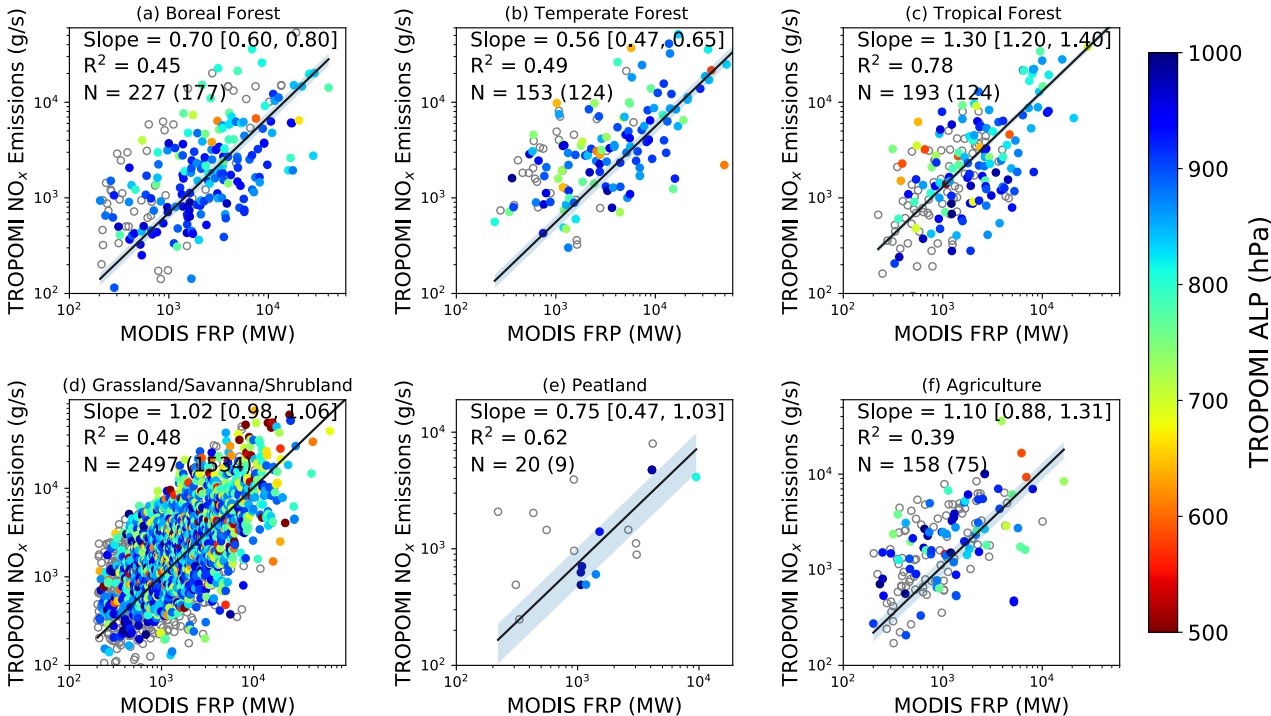

**Figure 3 Scatter plots between TROPOMI derived $NO_x$ emissions (g/s) and MODIS FRP (MW) for six fuel types: (a) boreal forest, (b) temperate forest, (c) tropical forest, (d) herbaceous fuel type that combines grassland, savanna and shrubland together, (e)**
**peatland, and (e) agricultural fires. The colours represent the TROPOMI ALP of the corresponding fire events. Fire events without**

valid ALP are shown as black circles. The black line indicates the regression line estimated from ordinary least squares with the intercept fixed at zero. The shadow represents the 95% CI of the fitted line, calculated based on the Student's t-distribution test. $R^2$ is the coefficient of the determination of the linear fit. N is the number of fire events, and the number of fires with valid TROPOMI ALH is in the parenthesis. Emissions and FRP are on log scales.

## 4.4 Comparison with previous studies

To compare with previous studies, we convert the ECs to emission factors (EFs) assuming a constant ratio of fuel consumption to FRP, $K_r = 0.41$ kg MJ$^{-1}$ (Vermote et al., 2009; Mebust et al., 2011; Mebust and Cohen, 2014). Figure 4 shows the TROPOMI derived NO$_x$ EFs and the associated uncertainties (95% CI) compared to previous studies. We find that satellite-derived NO$_x$ EFs (hereafter EFs$_{sat}$) are largely consistent with the mean reported in the Andreae (2019, hereafter EFs$_{andreae}$), which represent an up-to-date compilation of field and laboratory measurements over the last two decades. In most fuel types except for temperate forest, EFs$_{sat}$ are largely consistent with EFs$_{andreae}$ to within 30% difference. Our derived NO$_x$ EF$_{sat}$ for tropical forest (3.17 g/kg) is nearly twice as large as that in the boreal forest (1.70 g/kg), consistent with Andreae (2019), which also shows larger NO$_x$ EFs over the tropical forest (2.81 g/kg) than boreal forest (1.18 g/kg). However, for the temperate forest, the derived NO$_x$ EF$_{sat}$ (1.36 g/kg) is less than half of EF$_{andreae}$ (3.02 g/kg). There is a large spread of NO$_x$ EFs for the temperate forests in literature, ranging from 0.49 g/kg (Liu et al., 2017) to 7.44 g/kg (Yokelson et al., 2007). Our derived NO$_x$ EF$_{sat}$, however, is close to the in-situ estimates of NO$_x$ EFs (1.56 g/kg) from the recent aircraft campaign (i.e., WE-CAN) over the western US during summer 2018 (Lindaas et al., 2021). In non-forest fuel types, we find the smallest NO$_x$ EF$_{sat}$ (1.83 g/kg) over peatland, followed by grassland (2.49 g/kg), and agriculture (2.68 g/kg), which are close to EF$_{andreae}$. Using the standard TROPOMI NO$_2$ products without updating the *a priori* profile, the derived NO$_x$ EFs are 44 to 66% of EF$_{sat}$, and 26 to 68% of EFs$_{andreae}$. Assessment of TROPOMI NO$_2$ with *in situ* measurements also suggest TROPOMI NO$_2$ is biased low over polluted regions, and replacing the coarse-resolution *a priori* profile with fine-resolution simulations could largely reduce the low biases (Judd et al., 2020; Tack et al., 2021). Our derived NO$_x$ EFs are nearly 3 times larger than a previous study based on OMI observations, which suggest NO$_x$ EFs are lower than 1g/kg in all fuel types (Mebust and Cohen, 2014). Besides the differences in satellite instruments and methods, the discrepancy is partially due to less accurate representation of biomass burning emissions in the *a priori* profile of NO$_2$ in Mebust and Cohen (2014). Using the standard TROPOMI NO$_2$ products without updating the *a priori* profile, the derived NO$_x$ EFs are similar to those developed by Mebust and Cohen (2014) for boreal and temperate forest fires, but still higher over other fuel types.

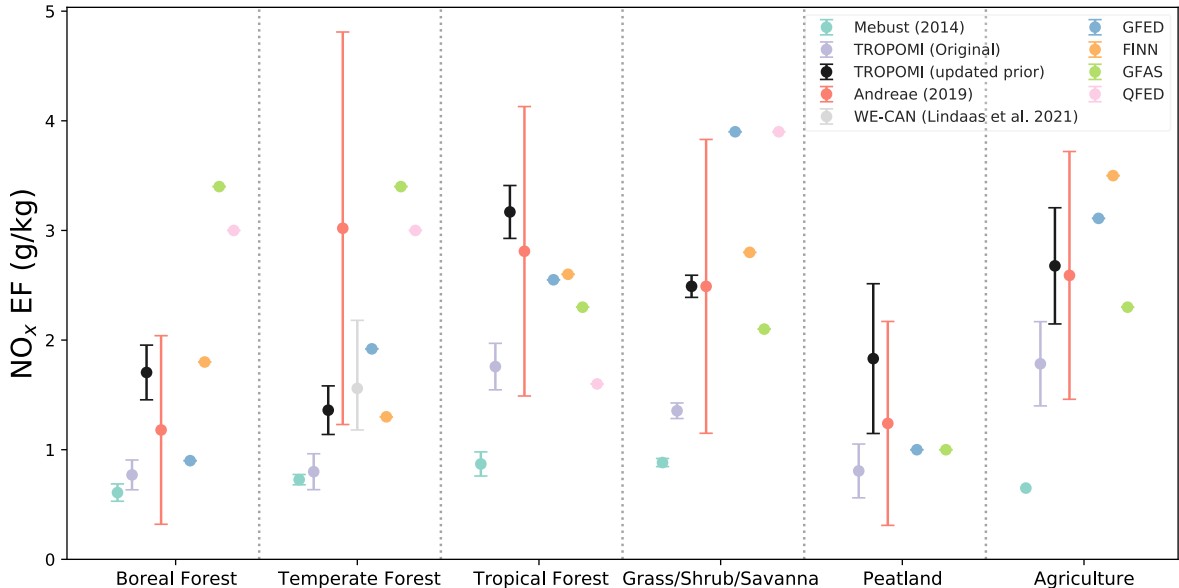

**Figure 4 Comparison of the TROPOMI-derived NOₓ emission factors with previous studies and those used in global biomass burning emission inventories. We include two estimates of NOₓ emission factors: one using the original TROPOMI NO₂ (purple); the other with updated the *a priori* profile for AMF calculation (black). The error bars of TROPOMI NOₓ EFs represent 95% CI calculated based on Student's t-distribution test. The red dots show the mean NOₓ EFs reported in previous studies compiled by Andreae (2019), and the error bars represent the standard deviation. The error bars of Mebust and Cohen (2014) are calculated using nonparametric bootstrap resampling. The error bars of Lindaas et al. (2021) indicate the overall uncertainty of measurements.**

As the development of biomass burning emission inventories is done by separate groups with different approaches, NOₓ EFs used in these inventories also differ. We compare our derived NOₓ EFs with those used in four commonly used global biomass burning emission inventories (Fig. 4), including: (1) Global Fire Emissions Database (GFED; van der Werf et al., 2017), (2) Fire Inventory from NCAR (FINN; Wiedinmyer et al., 2011), (3) GFAS (Kaiser et al., 2012), and (4) QFED (Darmenov and Silva, 2015). We find that satellite-derived EFsₛₐₜ best agree with those used in FINN for forest and grassland in terms of absolute magnitude and variations among fuel types. GFED and FINN use smaller EFs over boreal (0.9 g/kg and 1.8 g/kg) and temperate (1.9 g/kg and 1.3 g/kg) forest than tropical forest (2.6 k/kg), which is contrary to GFAS and QFED that use higher NOₓ EFs of 3.4 g/kg and 3.0 g/kg for the boreal and temperate forest than that for tropical forest (2.3 and 1.6 g/kg respectively). Our derived NOₓ EFₛₐₜ for tropical forest fires is larger than those used in emission inventories. For grassland, our derived NOₓ EFₛₐₜ of 2.49 g/kg is closest to that used in FINN (2.8 g/kg) and GFAS (2.1 g/kg), and smaller than that used in GFED and QFED (3.9 g/kg). For peatland fires, both GFED and GFAS use EF of 1.0 g/kg, which is smaller than our estimated EF of 1.83 g/kg, but we acknowledge a large uncertainty in our derived EF for peatland given the small number of samples. For agricultural fires, our derived NOₓ EFₛₐₜ (2.68 g/kg) is slightly higher than that used in GFAS (2.3 g/kg), but smaller than that used in GFED (3.11 g/kg) and FINN (3.5 g/kg).

### 4.5 Satellite-derived NOₓ lifetime and its driven factors

We find a large variation of $NO_x$ lifetimes in fire plumes. Figure 5 shows the variation of mean $NO_x$ lifetime as a function of $NO_x$ emissions at different wind speeds. We find an overall negative relationship between $NO_x$ lifetime and emissions: $NO_x$ lifetime decreases from over 5 hours for fires with emissions less than 500 g/s to less than 2 hours for fires higher than 5000 g/s (Fig. 5). We find similar $NO_x$ lifetime using original TROPOMI $NO_2$ data (Fig. S7), largely because the derived $NO_x$ lifetime is determined by the shape of fire plumes that are not affected by the *a priori*. The varying lifetime with emissions

suggests that the assumption of constant lifetime used in previous studies leads to an overestimate in emissions for small fires, while an underestimate of emissions for big fires. At low emission levels (< 2000 g/s), $NO_x$ lifetime tends to decrease with increasing wind speed, which is due to the vertical and horizontal diffusion effects that dilute the concentration of $NO_x$ and thus alters the rate of $NO_x$ removal due to the feedback on OH and the rate of $NO_x$ removal (Valin et al., 2013). However, as $NO_x$ emissions further increase (> 2000 g/s), the chemical removal becomes fast compared to dilution, and $NO_x$ lifetime no

longer depends on wind speed (Fig. 5). We note that the EMG derived $NO_x$ emissions depend on both $NO_x$ abundance and lifetime (Eq. (9)), and thus the observed negative emission-lifetime relationship may partially reflect that emissions are estimated from derived $NO_x$ lifetime. However, we find a similar negative relationship between $NO_x$ lifetime and TROPOMI $NO_2$ column density near the fire centre (Fig. S8), indicating that chemical feedback of $NO_x$ is the primary driver of the derived $NO_x$ lifetime.

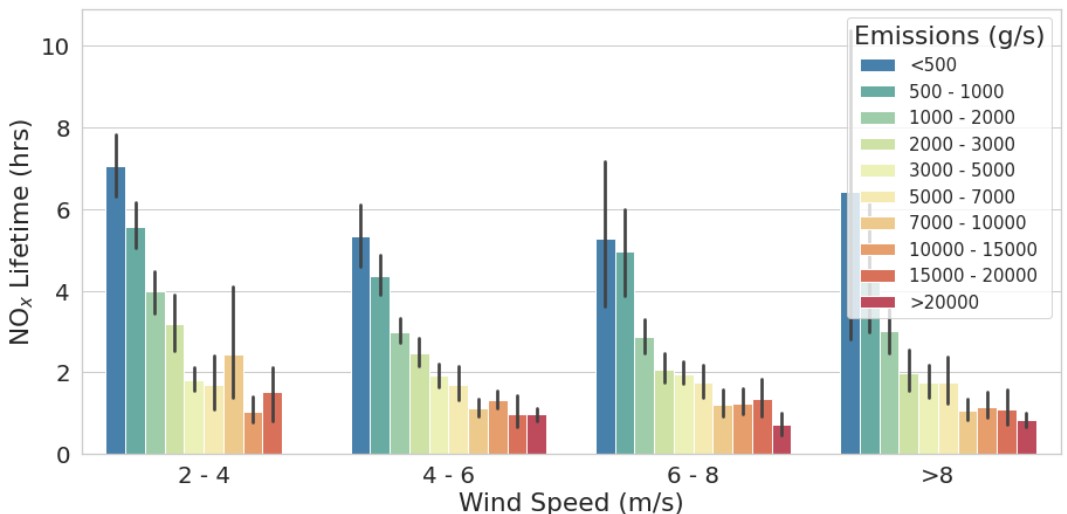


**Figure 5 The mean and standard deviation of TROPOMI derived NOₓ lifetimes from fires at different emissions (colour) and wind speeds. Fire episodes with less than 2 m/s wind speed are not shown.**

The $NO_x$ chemical lifetime, in theory, is determined by its loss to $HNO_3$ and $RONO_2$. We use the 1-D PECANS model to simulate $NO_x$ evolution downwind fire plumes and calculate a lifetime by fitting model simulated $NO_x$ concentration with the

EMG function (Eq. (7)). Figure 6 shows the dependence of the EMG fitted $NO_x$ lifetime on $NO_x$ emissions rate and P(HOₓ). At low $NO_x$ emissions, the $NO_x$ lifetime decreases rapidly with increasing $NO_x$ emissions, while almost independent of P(HOₓ),

indicating a NO$_x$-limited regime. At the NO$_x$-limited regime, increasing NO$_x$ facilitates the conversion from HO$_2$ to OH, and thus faster loss of NO$_x$ through formation of HNO$_3$ (e.g., Valin et al., 2014; Romer et al., 2020). The loss of NO$_x$ through formation of RONO$_2$ also increases with NO$_x$ emissions (e.g., Romer et al., 2020). As NO$_x$ emissions further increase, the NO$_x$ lifetime shows a strong dependence on P(HO$_x$), and the NO$_x$ lifetime increases slightly with NO emissions, indicating a NO$_x$-saturated regime. In the NO$_x$-saturated regime, as the loss of NO$_x$ through the formation of HNO$_3$ consumes both NO$_x$ and OH, increasing NO$_x$ leads to decreasing oxidative capacity and thus a longer NO$_x$ lifetime. If the NO$_x$ lifetime is driven entirely by changes in NO$_x$ concentration, the derived NO$_x$ lifetime should first decrease and then increase with NO$_x$ emissions, which is not found from the observed lifetime-emission relationship. Therefore, we infer that it is likely that P(HO$_x$) increases with fire intensity in fire plumes, which combined with increasing NO$_x$ abundances, leads to an overall decrease of NO$_x$ lifetime with NO$_x$ emissions. If we assume VOC reactivities and branching ratio $\alpha$ are fixed, we could use TROPOMI retrieved NO$_x$ emissions and lifetimes to infer an approximate level of P(HO$_x$). Figure 6 labels the satellite retrieved mean NO$_x$ lifetime at a given emission level, and the corresponding P(HO$_x$). To match the observed negative lifetime-emission relationship, P(HO$_x$) should also increase by near a factor of 4 from $15 \times 10^6$ molec/cm$^3$ for fires with lifetime longer than 4 hours to $60 \times 10^6$ molec/cm$^3$ for fires with lifetime smaller than 1 hour. The increase in P(HO$_x$) may be related to increasing emissions of HONO that generate OH. A recent study suggests that previously underestimated HONO emissions from fires are responsible for two-thirds of the HO$_x$ production from fresh fire plumes (Theys et al., 2020). The changes of P(HO$_x$), however, should be of secondary importance compared to NO$_x$ emissions in driving the observed variations in NO$_x$ lifetime, which can be evidenced from the slower rate of the increase in inferred P(HO$_x$) and small changes of the NO$_x$ lifetime at high NO$_x$ emissions (Fig. 5).

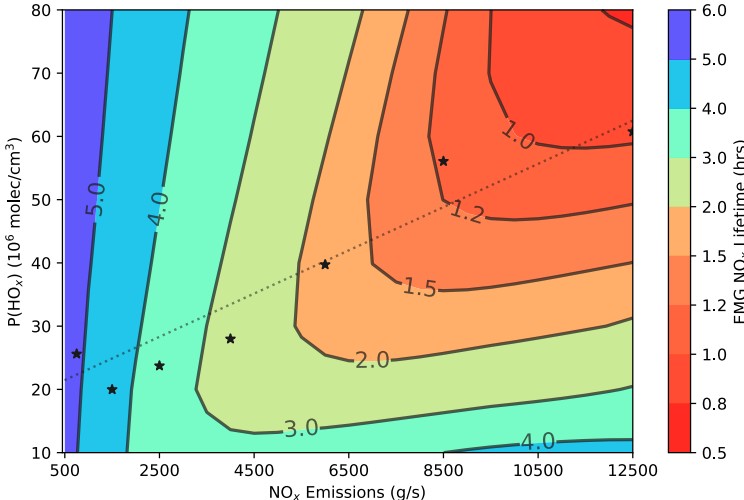

**Figure 6 Isopleth showing the NO$_x$ lifetime dependence on NO$_x$ emissions versus P(HO$_x$). We run the PECANS 1-D model with varying NO$_x$ emissions and P(HO$_x$). The lifetime is estimated by fitting the modelled NO$_x$ concentration along the wind direction with the EMG function (Eq. (7) and (8)). We run the model at a constant wind speed of 5 m/s, and the VOC reactivity is set constant at 4.8 s$^{-1}$ for both groups (RVOC and OVOC). The black stars show the TROPOMI observed mean NO$_x$ emissions and lifetime at wind speed between 4 to 6 m/s, where the P(HO$_x$) value is estimated as the level that gives the closest NO$_x$ lifetime as observations at given NO$_x$ emissions. The dashed line represents the fitted regression line.**

In addition to NO$_x$ emissions and P(HO$_x$), VOC reactivity is the third factor that affects the fitted NO$_x$ lifetime. At low NO$_x$ emissions, increasing RVOC reactivity leads to a shorter NO$_x$ lifetime, but its impacts become smaller and reverse with increasing NO$_x$ emissions (Fig. 7a). At low NO$_x$ emissions, increasing RVOC reactivity facilitates the loss of NO$_x$ through the formation of RONO$_2$. At high NO$_x$ emissions, as both RVOC and NO$_x$ consumes OH, increasing VOC leads to a longer NO$_x$ lifetime. The formation of PAN acts as a temporal reservoir of NO$_x$, which also affects the evolution of NO$_x$. The formation of PAN depends on the concentration of OVOCs, OH level and temperature. Figure 7b shows the dependence of EMG fitted NO$_x$ lifetime as a function of NO$_x$ emission rate and the reactivity of PAN's immediate precursor (OVOC). We find that increasing PAN formation through increasing OVOC reactivity will lead to an overall increase in EMG fitted NO$_x$ lifetime. The impact of OVOC is especially evident at low levels of NO$_x$ emissions (Fig. 7b). Without PAN formation, the fitted NO$_x$ lifetime will be shorter than that derived from TROPOMI observations, suggesting that PAN formation plays a non-negligible role in determining the evolution of NO$_x$ and also the effective lifetime of NO$_x$.

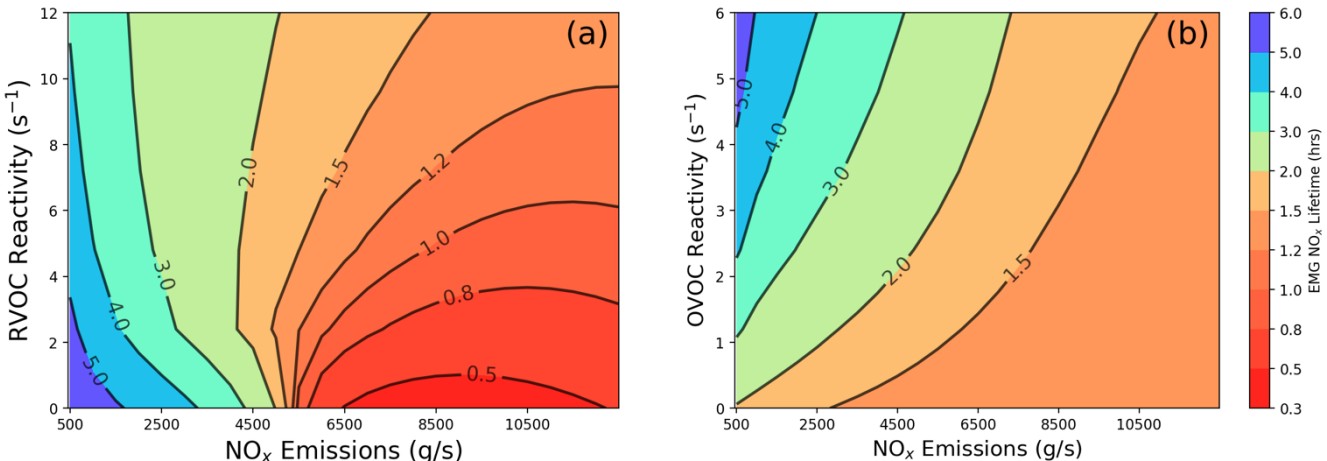

**Figure 7 Same as Fig. 6 but showing the NO$_x$ lifetime dependence on NO$_x$ emissions versus the reactivities of (a) RVOC and (b) OVOC. We run the model at a constant wind speed of 5 m/s, and the P(HO$_x$) is set constant at 50 × 10$^6$ molecules/cm$^3$.**

## 5 Discussions on uncertainties

### 5.1 Uncertainties of satellite retrieved NO$_2$ columns

Satellite retrievals of NO$_2$ columns are subject to uncertainties with spectral fitting, and uncertainties from the *a priori* NO$_2$ profile shape and scattering weights needed for calculating AMFs and stratospheric NO$_2$ columns. Boersma et al. (2018) suggest an overall uncertainty of 35% to 45% for single-pixel OMI NO$_2$ retrieval, which should be smaller for TROPOMI NO$_2$ given its improved signal-to-noise ratio (van Geffen et al., 2020; Verhoelst et al., 2021). Due to the sporadic nature of fires, there is no validation of TROPOMI NO$_2$ columns with fire NO$_x$ with ground-based measurements. Validation with ground-based Differential Optical Absorption Spectroscopy (DOAS) and Pandora measurements in urban stations show an overall

good agreement between TROPOMI retrieved and ground-based $NO_2$ columns, and an overall negative bias is reported, but the biases vary with stations (Lambert et al., 2020; Verhoelst et al., 2021).

Over polluted regions, the uncertainty and bias of single-pixel satellite retrieval of $NO_2$ columns is dominated by uncertainties of the AMF (Boersma et al., 2018). We replace the *a priori* profile shape of $NO_2$ using NASA GEOS-CF simulations that include daily biomass burning emissions, which leads to higher $NO_x$ emissions factors that are more consistent with in situ measurements. Bousserez (2014) similarly suggests that using a fire-specific $NO_2$ profile shape can lead to a near 60% reduction in AMF. In the NASA GEOS-CF, 65% biomass burning emissions are distributed within the boundary layer, which will lead to negative biases in AMF for large fires with plumes that rise well above the boundary layer. In our study, since the ALH is below 2000m for the majority of fires, model uncertainties of the injection height should have small impacts on the retrieval of $NO_2$ columns. However, we notice that fire events with low ALP (< 700 hPa) tend to show higher $NO_x$ emissions per FRP, and most outliers in Fig. 3 are associated with high aerosol layers. Since satellite instruments are more sensitive to $NO_2$ at higher altitudes, the inaccurate assumption of $NO_2$ concentrated within the boundary layer will lead to an underestimate of the AMF and thus an overestimate of the retrieved vertical $NO_2$ columns if the majority of $NO_2$ is injected to the free troposphere.

In addition, the large amounts of aerosols emitted from wildfires may also impact the retrieval of $NO_2$ columns from space. The impacts of aerosols depend on the magnitude, location, and optical properties (absorbing vs. scattering) of aerosols (Bousserez, 2014; Lin et al., 2015). In the TROPOMI $NO_2$ retrieval algorithm, the effects of aerosols are implicitly accounted for through modifying the cloud properties (Boersma et al., 2011). Bousserez (2014) suggests an explicit correction is needed in the presence of clouds and scattering aerosols, and the effects of aerosol correction can be as large as 100% when cloud fraction is 30% and AOD is higher than 1. In our study, the mean cloud fraction of the selected fire plumes is 9%, and the mean AOD is 0.22, corresponding to a small uncertainty of less than 20% based on Bousserez (2014).

## 5.2 Uncertainties of MODIS FRP

Since we derive $NO_x$ EFs from the linear regression between TROPOMI $NO_x$ emissions and MODIS FRP, uncertainties in MODIS FRP should also contribute to uncertainties of the satellite derived $NO_x$ EFs. Detection of MODIS FRP may be obscured by cloud, aerosols or canopy cover. The uncertainty of FRP, however, is lower than 5% for fires that aggregate 30 or more active fire pixels together (Freeborn et al., 2014). Our selected fire events are aggregated by 37 active fire pixels on average. To evaluate if our results are robust with the choice of FRP data, we conduct similar analysis with FRP measurements from Suomi NPP VIIRS sensors. In general, MODIS and VIIRS FRP are in good agreement ($R^2 = 0.72$, Fig. S9). VIIRS FRP is lower than MODIS FRP by 5.8% on average. Deriving $NO_x$ emission factors using VIIRS FRP, we find a slight increase of $NO_x$ EFs for forest and agricultural fires, but a decrease for peatland (Fig. S10). For herbaceous fires, where a large number of fires are sampled, the derived $NO_x$ EF remains almost unchanged, suggesting that the difference in MODIS and VIIRS FRP should diminish as we increase the sample size. The overall relationship between $NO_x$ emissions and FRP is similar for both VIIRS and MODIS, though stronger correlation is found for MODIS FRP. Overall, we estimate the difference in $NO_x$ EFs

using MODIS versus VIIRS FRP is ~20%. Here we assume linear relationship between emission rate and FRP. While the validity of the relationship has been verified in laboratory (Wooster et al., 2005; Freeborn et al., 2008), field experiments (Wiggins et al., 2020) and satellite observations (Ichoku and Kaufman, 2005), the choice of the mass-to-energy conversion factor ($K_r$) slightly differ in Wooster et al. (2005, 0.368 g/MJ) and in Freeborn et al. (2008, 0.453g/MJ), suggesting an uncertainty of order 10% for the value of $K_r$.

## 5.3 Uncertainties in the EMG approach

The first step of the EMG approach is to rotate TROPOMI observations along the wind direction. The derived $NO_x$ lifetimes and emissions are therefore subject to uncertainties of the wind direction and speed due to uncertain plume heights that cross wind shear, or the plume thermodynamics that are not captured by ERA5 wind data. In the case the fire plume does not align with wind direction, calculating line density along the wind direction should lead to an underestimate of the *e*-folding distance. In this study, we only select fires with less than 30˚ rotation bias, and the mean rotation bias is 10˚, which corresponds to less than 2% underestimate of the e-folding distance. Uncertainty and variance of the wind speed, however, should lead to errors in the derived $NO_x$ lifetimes. Here we determine the wind speed by interpolating the wind profile to TROPOMI derived aerosol layer. Comparison of TROPOMI ALH with plume height from the Cloud-Aerosol LIdar with Orthogonal Polarization (CALIOP) and the Multi-angle Imaging SpectroRadiometer (MISR) measurements suggest that TROPOMI ALH is overall 500 m lower (Griffin et al., 2020; Nanda et al., 2020). We estimate that an increase of 500 m ALH corresponds to ~22% increase of the wind speed on average, meaning that $NO_x$ lifetime will decrease by ~18%.

Here we use the EMG approach to derive an effective $NO_x$ lifetime of the entire fire plume. Chemical nonlinearities can result in an effective chemical lifetime that is averaged over the plume where at each point in the plume evolution a different chemical lifetime occurs. Besides, the effective lifetime in practice will be confounded by the mixing such as those plume movement in different directions that reduces the line density. To assess if EMG fitted effective $NO_x$ lifetime is indicative of the chemical lifetime, we use the PECANS model to calculate an EMG fitted lifetime and a chemical lifetime of $NO_x$ from two permanent losses of $NO_x$ through the formation of $HNO_3$ and $RONO_2$ over downwind region (*i.e.*, mean $NO_x$ concentration divided by the mean chemical loss of $NO_x$). The chemical lifetime of $NO_x$ varies with location (Fig. 8a), which reaches a minimum near the source centre at low $NO_x$ emissions ($NO_x$-limited regime), but shows maximum at high $NO_x$ emissions ($NO_x$-saturated regime). The effect of varying lifetime on the emission estimates is not considered with the EMG approach, which instead gives an overall effective $NO_x$-lifetime of the plume. At low $NO_x$, the EMG fitted lifetime is higher than the chemical lifetime at the fire centre, but the reverse occurs at high $NO_x$ (Fig. 8a). We notice that the EMG fitted lifetime is overall more consistent with the chemical lifetime around 40 to 50 km downwind from the fire centre, which should vary depending on the model input. Figure 8b compares the EMG fitted lifetime with overall chemical lifetime over downwind regions under different $NO_x$ emissions, P(HO$_x$) and VOC reactivities. We find a moderate correlation between EMG fitted lifetime and chemical lifetime ($R^2 = 0.37$). If PAN formation is not included, a nearly perfect correlation is found between the EMG fitted and chemical lifetime ($R^2 = 0.93$). In the lower troposphere, PAN is generally considered as a sink of $NO_x$ near the fire, but a source of $NO_x$

over downwind region, which deepens the gradient of $NO_2$ near the source but flattens the gradient downwind (Valin et al., 2013). For those large biomass burning events that inject PAN into upper troposphere, PAN acts as a stable reservoir of $NO_x$, leading to long-range transport of $NO_x$ (Tereszchuk et al., 2013). The EMG approach is unable to capture the effects of PAN formation on the evolution of $NO_x$ as it assumes $NO_x$ decays exponentially. In the presence of PAN formation, we find the EMG approach tends to overestimate $NO_x$ lifetime at low $NO_x$ emission (< 5000 g/s), in which the flattening effect of PAN is more evident, while underestimating $NO_x$ lifetime at high $NO_x$ emissions and low $P(HO_x)$, in which the deepening effect of PAN takes over. Overall, we assess that the overestimate at low $NO_x$ emissions (< 5000 g/s) cause around 33% negative biases to the derived emissions, while 18% positive biases at high $NO_x$ emissions (> 5000 g/s). In-situ measurements show rapid formation of PAN in young smoke within 4 hours of aging, and PAN contributes about 25% of the total reactive nitrogen (Alvarado et al., 2010; Juncosa Calahorrano et al., 2021), suggesting a non-negligible role of PAN as a sink of $NO_x$ near fire source.

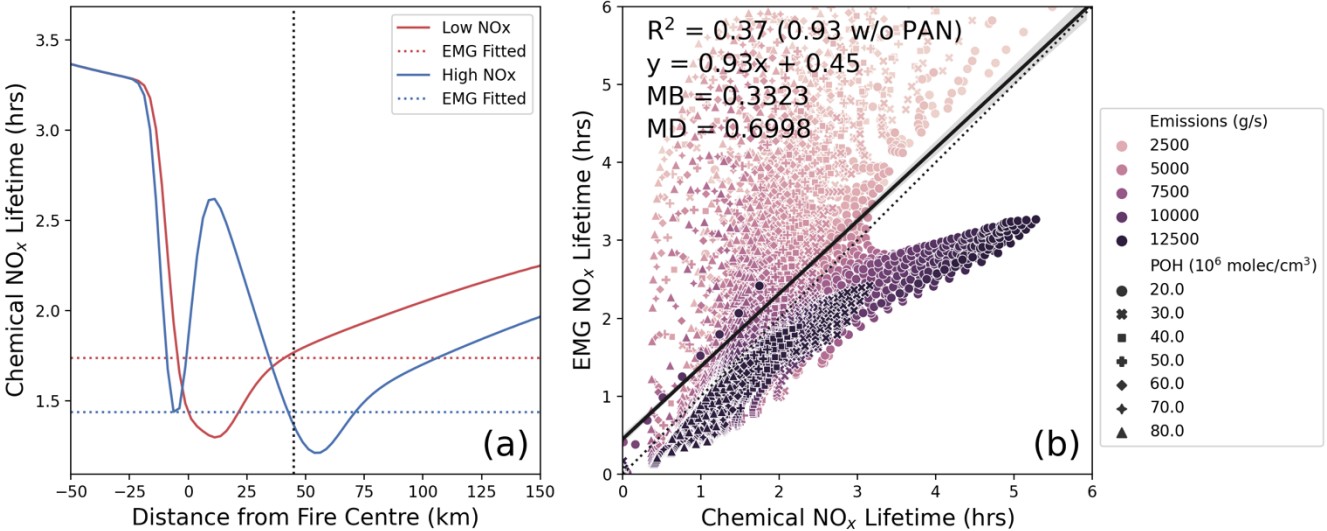

Figure 8 (a) Illustration of the variation of $NO_x$ chemical lifetime with distance from fire centre at both low (red) and high $NO_x$ emissions (blue). The EMG fitted effective lifetime is shown as horizontal dash lines. The vertical line indicates the distance where chemical lifetime and EMG fitted lifetime agree. (b) Scatter plot between EMG fitted versus overall chemical $NO_x$ lifetime downwind calculated using varying $NO_x$ emissions, $P(HO_x)$ and VOC reactivities, where the colours represent different emission levels, and the symbols represent different levels of $P(HO_x)$. The chemical lifetime is calculated as the mean $NO_x$ concentration divided by the mean chemical loss of $NO_x$ through the formation of $HNO_3$ and $RONO_2$ over downwind region. We define the regional mean as the region between fire centre and the downwind area where $NO_x$ concentration is higher than background, where background value is estimated from the EMG function (B in Eq. (7)).

## 6 Conclusions

We estimate $NO_x$ emissions and lifetimes from over 3000 fires globally using daily TROPOMI retrievals of $NO_2$ tropospheric columns, and derive $NO_x$ emission factors by linking TROPOMI derived $NO_x$ emissions with MODIS FRP. The overall derived $NO_x$ emissions factors are 1.70 g/kg, 1.36 g/kg and 3.17 g/kg for the boreal, temperate and tropical forest, 2.49 g/kg

for herbaceous (grassland, savannas and shrubland combined) fires, 1.83 g/kg for peatland, and 2.67 g/kg for agricultural

burning. Satellite-derived $NO_x$ emission factors are largely consistent with the mean $NO_x$ emission factors reported by previous field studies (Andreae, 2019). By studying a large number of fires globally, we provide more representative $NO_x$ emission factors. These top-down emission estimates of $NO_x$ could be used to assess biomass burning emission inventories in terms of both emission factors and fuel consumption, which could help diagnose the causes of discrepancies among different emission inventories.

Our study features three improvements over previous studies that use satellite measurements to derive $NO_x$ emissions (Mebust et al., 2011; Mebust and Cohen, 2014; Schreier et al., 2015). First, we use observations from TROPOMI with finer spatial resolution and improved signal-to-noise ratio. Second, we relax the assumptions of constant $NO_x$ lifetime by directly estimating lifetime through fitting the evolution of $NO_x$ downwind with the EMG approach. Third, we replace the *a priori* profile of TROPOMI $NO_2$ retrieval with a high-resolution global model simulation from NASA GEOS-CF simulations to calculate AMF.

This update result in steeper gradients between the plumes and the background, and more accurate description of $NO_2$ vertical shape, reducing the discrepancy between satellite and in-situ derived estimates of $NO_x$ emission factors. The resolution of current global model simulation, however, is not sufficient to resolve the fine-scale chemical evolution of fire plumes, and better treatment of the fire injection is needed (Paugam et al., 2016). Assessment of the satellite retrieval uncertainty will benefit from high-resolution regional simulations combined with *in situ* measurements that sample individual fire smokes from

the point of emission to downwind regions (Juncosa Clahorrano et al., 2021; Lindaas et al., 2020).

We observe decreasing $NO_x$ lifetime with increasing fire $NO_x$ emissions, which is indicative of $NO_x$-limited chemistry, where increasing $NO_x$ emissions makes the chemical loss of $NO_x$ more efficient. However, for the largest fires with high $NO_x$, a regime transition from a $NO_x$-limited to $NO_x$-saturated regime is expected, where increasing $NO_x$ emissions leads to a longer $NO_x$ lifetime. Using a 1-D idealized plume model to interpret the factors affecting the $NO_x$ lifetime, we infer that $P(HO_x)$ must

also increase with fire intensity, consistent with observations that indicate a large source of HONO in fires (Theys et al., 2020; Peng et al., 2020). The formation of PAN also impacts the $NO_x$ lifetime, but the evolution of $NO_x$ due to the formation of PAN and its thermal decomposition over downwind areas is not well captured by the EMG approach that assumes exponential decay with $NO_x$ downwind. Future studies will benefit from the integrative analysis of satellite retrievals of $NO_2$, HONO and PAN to more completely describe the chemical evolution of reactive nitrogen from wildfires, thus allowing for better prediction of

the air quality impacts of fires. TROPOMI is limited to single overpass per day, which cannot resolve the short-term evolution of fire plumes observed by *in situ* measurements (e.g., Juncosa Calahorrano et al., 2021). The newly launched or upcoming geostationary satellite instruments such as GEMS and TEMPO will offer an unprecedented opportunity to continuously observe the emissions and chemical evolution of $NO_x$ from fires that will no longer be limited to a single snapshot (Chance et al., 2013; Kim et al., 2019).

**Data and code availability:**

TROPOMI NO$_2$ data (doi:10.5270/S5P-s4ljg54) and TROPOMI ALH (doi:10.5270/S5P-j7aj4gr) are available from NASA Goddard Earth Sciences (GES) Data and Information Services Center (DISC, https://disc.gsfc.nasa.gov/datasets/). MODIS and VIIRS FRP data are available from NASA Earth Data Fire Information for Resource Management Systems (https://earthdata.nasa.gov/earth-observation-data/near-real-time/firms). MAIAC AOD data are available from NASA's Land Processes Distributed Active Archive Center (LP DAAC) located at the USGS Earth Resources Observation and Science (EROS) Center (https://e4ftl01.cr.usgs.gov/MOTA/MCD19A2.006/). ERA5 hourly wind data are available from the Copernicus Climate Service (C3S) Climate Data Store (https://cds.climate.copernicus.eu/cdsapp#!/dataset/reanalysis-era5-pressure-levels). GEOS-CF simulations are available from NASA Global Modeling and Assimilation Office (https://gmao.gsfc.nasa.gov/weather_prediction/GEOS-CF/data_access/). PECANS code is available from https://github.com/joshua-laughner/PECANS .

**Acknowledgments**:

X. Jin is supported by the NOAA Climate and Global Change Postdoctoral Fellowship Program, administered by UCAR's Cooperative Programs for the Advancement of Earth System Science (CPAESS) under award # NA18NWS4620043B. This work is also supported by NASA Grant 80NSSC18K0624.

**Author contribution**:

XJ and RC conceived the project. QZ provided the PECANS code. XJ carried out the data analysis and prepared the manuscript with contributions from all co-authors.

**Competing interests:**

The authors declare that they have no conflict of interest.

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
