# Peer review of "Direct estimates of biomass burning $NO_x$ emissions and lifetimes using daily observations from TROPOMI"

_Atmospheric Chemistry and Physics, 2021_

## Referee Comment (RC1)

Title: Direct estimates of biomass burning NOx emissions and lifetime using daily observations from TROPOMI
Author(s): Xiaomeng Jin et al.
MS No.: acp-2021-381
MS type: Research article

General comments:

This manuscript show the capabilities of NO2 satellite retrievals from the TROPOspheric Monitoring Instrument from wildfire smoke plumes. An exponentially modified Gaussian (EMG) approach, NO2 TROPOMI retrievals, MODIS FRP, aerosol layer heights from TROPOMI, and reanalysis data are used to estimate the fire emissions of NO2 and its lifetime in the smoke plumes from >3000 fires globally. Fire locations and intensity are derived using MODIS Fire Radiative Power (FRP). The authors used GEOS-CF to modify the NO2 a-priori profile to correct the low bias of NO2 retrievals over fire plumes. The authors made a detailed comparison of their results with previous references regarding biomass burning emissions. Finally, the authors found that there is an anticorrelation between fire size and NO2 lifetime, possibly attributed to the higher emissions of HOx in larger fires. The manuscript has detailed methods, has an in-depth discussion of uncertainties, results are interesting and presented in a clear way, and is well written. Therefore, I suggest publication after minor revisions.

Specific comments
L66: looks like a good place to introduce PAN and organic nitrates?
L83 and elsewhere: It would be good to specify the local time of TROPOMI overpass. I am guessing that this influenced the choice of the NOx/NO2 ratio. I would like a little more discussion about how this ratio can change as time of the day. Is there any variation of it as the plume ages?
L153. I had a little bit of trouble understanding if the calculated lifetime included dilution. Reading the manuscript a second time, it was clearer. Is there a way that you can separate dilution from chemistry (i.e., using a NO2 normalized excess mixing ratio with respect to CO or CO2?). This might help you to further constrain or interpret your idealized plume model.
L175 Repeated comment but I would like to see more discussion about NOx/NO2 ratio as a function of the time of the day or distance from plume. References that might be helpful include Yokelson et al., 2009; Akagi et al 2011; Alvarado et al 2010; Juncosa Calahorrano 2021).
L195 Section 3.4 Did you used a plume specific NO2 background (i.e., for the background condition the day the plume was retrieved by TROPOMI)? For locations with fire seasons that last for months, background conditions can change because of the presence of dilute smoke in the area.
L220. Please include more details on figure S1 e.g., rotation angle. Might be good to have a final figure after rotation as well
L221 Can you please explain how you differentiate between the fire center vs. the apparent fire center? Is the first based on the FRP and the latter based on visual inspection? Also, this line needs more detail. What do you mean by "give good fitting statistics"? What is the correlation that needs to satisfy R2>0.5?
Figure 1. I would have expected more tropical fires detected by TROPOMI. Did one of the criteria to remove plumes excluded those? Why? Also, it looks that there are not many fires in the equatorial line in South America and Africa, which is odd. Can you explain please? Perhaps this is an issue with the satellite retrievals?

L235 How did you identify Ag fires?

L247 and Figure 1 Very nice section. I was just wondering why you didn't remove the small fire towards the upper left side of the bigger fire. I assumed your criteria will remove it because it is clearly overlapping with the bigger one. Please explain.

L267 This sentence is confusing. I would remove the first part (before the comma) from this sentence.

L284 I am a little concerned about the conclusion that emissions and Fire Radiative Power correlate. In past field campaign, its being shown that reduced vs. oxidized emissions of nitrogen correlate very well with the Modify Combustion Efficiency (MCE) (i.e., smoldering vs. flaming) but it has been difficult to correlate them directly to FRP. The fire condition can also impact the chemistry in PECANS. If the fire has lower MCEs, there will be a lot of reduced nitrogen (e.g., NH3) compared to oxidized nitrogen (e.g., HONO that produces OH, NOx, etc). I know that getting MCE from many different fires is challenging, but I am not convinced that using FRP is the right approach. At the very least, there should be more discussion about how MCE affects emissions in the manuscript.

L333 Here is where I realized that the lifetime included dilution. Please include a few sentences somewhere earlier in the manuscript identifying all the loss processes that the estimated NO2 lifetime includes.

L465 I think you should discuss the thermal dependency of PAN. If the plume is injected high enough, PAN can be stable and thus its transportation can be very efficient (e.g., not a source of NOx close to the plume, at least in the time scales this relevant to this manuscript).

L468 We know well that PAN forms rapidly in fire smoke plumes and that its production plateaus after ~4 after of plume aging (Yokelson et al., 2009, Akagi et al., 2011, Alvarado et al., 2010, Juncosa Calahorrao 2021). It might be helpful to look at how the ratio of PAN/NOx changes as the plume ages and have some discussion about it.

L470 can you find another word for "estimate" so it is not right before "overestimate"

Great paper!

References

Yokelson, R. J., Crounse, J. D., DeCarlo, P. F., Karl, T., Urbanski, S., Atlas, E., Campos, T., Shinozuka, Y., Kapustin, V., Clarke, A. D., Weinheimer, A., Knapp, D. J., Montzka, D. D., Holloway, J., Weibring, P., Flocke, F., Zheng, W., Toohey, D., Wennberg, P. O., … Shetter, R. (2009). Emissions from biomass burning in the Yucatan. *Atmos. Chem. Phys.*, 28.

Akagi, S. K., Craven, J. S., Taylor, J. W., McMeeking, G. R., Yokelson, R. J., Burling, I. R., Urbanski, S. P., Wold, C. E., Seinfeld, J. H., Coe, H., Alvarado, M. J., & Weise, D. R. (2012). Evolution of trace gases and particles emitted by a chaparral fire in California. *Atmospheric Chemistry and Physics*, *12*(3), 1397–1421. https://doi.org/10.5194/acp-12-1397-2012

Alvarado, M. J., Logan, J. A., Mao, J., Apel, E., Riemer, D., Blake, D., Cohen, R. C., Min, K.-E., Perring, A. E., Browne, E. C., Wooldridge, P. J., Diskin, G. S., Sachse, G. W., Fuelberg, H., Sessions, W. R., Harrigan, D. L., Huey, G., Liao, J., Case-Hanks, A., … Le Sager, P. (2010). Nitrogen oxides and PAN in plumes from boreal fires during ARCTAS-B and their impact on ozone: An integrated analysis of aircraft and satellite observations. *Atmospheric Chemistry and Physics*, *10*(20), 9739–9760. https://doi.org/10.5194/acp-10-9739-2010

Juncosa Calahorrano, J. F., Lindaas, J., O'Dell, K., Palm, B. B., Peng, Q., Flocke, F., Pollack, I. B., Garofalo, L. A., Farmer, D. K., Pierce, J. R., Collett, J. L., Weinheimer, A., Campos, T., Hornbrook, R. S., Hall, S. R., Ullmann, K., Pothier, M. A., Apel, E. C., Permar, W., … Fischer,

E. V. (2021). Daytime Oxidized Reactive Nitrogen Partitioning in Western U.S. Wildfire Smoke Plumes. *Journal of Geophysical Research: Atmospheres*, *126*(4). https://doi.org/10.1029/2020JD033484

---

## Referee Comment (RC3)

The authors directly estimate $NO_X$ emissions and lifetime for fires by using an exponentially modified Gaussian analysis of tropospheric $NO_2$ columns observed by TROPOMI. The authors firstly correct the low bias of TROPOMI retrieved NO2 columns by replacing the *a priori* profile of NO2 with the GEOS-CF simulated profile at a finer resolution of 0.25. Representative NOx emission factors for six fuel types are derived by using the observations of fire radiative power from MODIS. The authors also discussed the uncertainties and capabilities of the method thoroughly. The scope fits ACP and the scientific idea is new. I recommend the paper be published after the authors address the following comments.

Major comments:

1, A better result is expected after the authors used *a priori* $NO_2$ profile with 0.25 to replace the one used by the operational product. However, this spatial resolution is still much coarser than TROPOMI's, which implies that nearby pixels use the same profile shape. As the authors presented in the paper that fire events normally take place locally. How much uncertainty can it contribute?

2, This plume-based method works only when the wind speed is not small, that is the plume exists. The authors keep every case even with very low wind speeds (< 2m/s). Should these cases be removed for the scientific reason?

3, The authors argue that the difference between TROPOMI and OMI derived is mainly due to the *a priori* $NO_2$ profiles, which is not accurate. The VCDs of TROPOMI are found to be lower than OMI's over many places, which is mainly caused by impropriate surface albedos or cloud pressures. Please give more solid proof if the authors want to draw this conclusion (i.e. section 4.4).

4, Section 3.4 is long and complicated. A flowchart is helpful to explain the procedure or moving this part to the supplementary.

5, The authours intend to derive representative $NO_X$ emission factors for six fuel types. However, satellite observations are available once per day, and some fire events can last for several days. In these cases, the fire intensity and the chemical condition also change. The authours, at least, should give an example to explain how to consider the emission factor for a certain fuel type.

Specific comments:

1. Line 11: "behaviour" should be "behavior". "occur" should be "that occur".

2. Line 15: The sentence is a little confusing. I think the authors recalculate the $NO_2$ VCD of every pixel with the GEOS-CF simulated profile not only over the fire plumes?

3. Line 24 and 27: Please list enough examples when you give examples.

4. Line 42: "the fire detection".

5. Line 44: "has a finer spatial resolution".

6. Line 46: "spatial resolutions"

7. Line 57 and 64: Please cite recent and more relevant studies about TROPOMI.

8. Line 83: "afternoon global" is quite obscure, please specify the overpassing time is around 13:30 local time.

9. Line 86-93: Do you use the S5P operational product that retrieved by KNMI? If so, you should cite van Geffen et al., (2019) when introducing the way of retrieval. Besides, you should also cite the validation paper (i.e. Tijl et al., 2021 https://amt.copernicus.org/articles/14/481/2021/ ) when discussing the underestimation of S5P.

   van Geffen, J. H. G. M., Eskes, H. J., Boersma, K. F., Maasakkers, J. D., and Veefkind, J. P.: TROPOMI ATBD of the total and tropo spheric NO2 data products (issue 1.4.0), Royal Netherlands Meteorological Institute (KNMI), De Bilt, the Netherlands, 2019.

10. Line 165: The format of the reference is "Laughner and Cohen, (2019)".

11. Line 203: Not very clear to me why "3 to 30 days before and after the fire day" is 56 days in total.

12. Line 206: "filter" should be "filters".

13. Line 224-225: Please give specific examples to explain "We also exclude fires in which TROPOMI NO2 line densities are monotonically increasing or decreasing within the region."

14. Line 241: "10000 MW" should be "10,000 MW"

15. The resolution of Figure 2 is too low. It's better to start with the original TROPOMI $NO_2$ data before (a).

---

## Author Response (AR1)

**Reply to Reviewer 1**

*This manuscript show the capabilities of NO2 satellite retrievals from the TROPOspheric Monitoring Instrument from wildfire smoke plumes. An exponentially modified Gaussian (EMG) approach, NO2 TROPOMI retrievals, MODIS FRP, aerosol layer heights from TROPOMI, and reanalysis data are used to estimate the fire emissions of NO2 and its lifetime in the smoke plumes from >3000 fires globally. Fire locations and intensity are derived using MODIS Fire Radiative Power (FRP). The authors used GEOS-CF to modify the NO2 a-priori profile to correct the low bias of NO2 retrievals over fire plumes. The authors made a detailed comparison of their results with previous references regarding biomass burning emissions. Finally, the authors found that there is an anticorrelation between fire size and NO2 lifetime, possibly attributed to the higher emissions of HOx in larger fires. The manuscript has detailed methods, has an in-depth discussion of uncertainties, results are interesting and presented in a clear way, and is well written. Therefore, I suggest publication after minor revisions.*

Reply: We would like to thank the reviewer for their constructive feedback and time spent reviewing this paper. Below is our response to reviewer's comments.

**Specific comments.**
*1. L66: looks like a good place to introduce PAN and organic nitrates?*

Reply: Great suggestion.We now revise the sentence as follows:

Satellite instruments observe fire $NO_x$ plumes as a mixture of fresh and aged smoke. $NO_x$ is a short-lived species, and its concentration will decay in the plume due to the formation of nitric acid ($HNO_3$), peroxyacetyl nitrate (PAN) and organic nitrates ($RONO_2$).

*2. L83 and elsewhere: It would be good to specify the local time of TROPOMI overpass. I am guessing that this influenced the choice of the NOx/NO2 ratio. I would like a little more discussion about how this ratio can change as time of the day. Is there any variation of it as the plume ages?*

Reply: The overpass time of TROPOMI is around 1:30 PM local time. We have revised the sentence as follows:

TROPOMI provides afternoon (~ 1:30 PM local time) global observations in the UV−visible−near infrared−shortwave spectra with a fine spatial resolution of $7 \times 3.5$ km$^2$ at nadir (increased to $5.5 \times 3.5$ km$^2$ since August 2019).

We use a constant $NO_x/NO_2$ ratio of 1.32, which is in between the measured mean $NO_x/NO_2$ ratio of 1.50 reported in Akagi et al. (2012) and 1.24 in Juncosa Clahorrano et al. (2021). Juncosa Clahorrano et al. (2021) show the ratio the median $NO_x/NO_2$ varies little from fire centre to plume edge. Juncosa Clahorrano et al. (2021) show the ratio peaks at noon, and our

chosen ratio is indeed at the upper bound of their reported ratio. We discuss the choice of $NO_x/NO_2$ in the revised manuscript as follows:

$\gamma$ is assumed to be 1.32, which is in between measured mean $NO_x/NO_2$ ratio of 1.50 reported in Akagi et al. (2012) and 1.24 in Juncosa Clahorrano et al. (2021). We assume a constant $\gamma$ because $O_3$ and the photolysis rate of $NO_2$ varies little in the plume, and the time scale for NO and $NO_2$ to reach steady state is of order 100s (Alvarado and Prinn, 2009). Juncosa Clahorrano et al. (2021) shows the $NO_x/NO_2$ ranges from 1.15 to 1.50 near the fire centre before 3PM LST, but the median $NO_x/NO_2$ varies little from centre to plume edge. Mebust et al. (2011) suggest the uncertainty of $NO_x/NO_2$ is ~20%.

*3. L153. I had a little bit of trouble understanding if the calculated lifetime included dilution. Reading the manuscript a second time, it was clearer. Is there a way that you can separate dilution from chemistry (i.e., using a NO2 normalized excess mixing ratio with respect to CO or CO2?). This might help you to further constrain or interpret your idealized plume model.*

Reply: We use a Gaussian function in Eq. (7) to represent the smoothing in line densities due to dilution. The calculated lifetime should be representative of chemical lifetime if the transport speed is uniform, the direction is constant and deposition is negligible (De Foy et al., 2014). We have clarified this:

The effective lifetime should represent chemical lifetime of $NO_x$ if the transport speed is uniform, the direction is constant and deposition is negligible (De Foy et al., 2014).

*4. L175 Repeated comment but I would like to see more discussion about NOx/NO2 ratio as a function of the time of the day or distance from plume. References that might be helpful include Yokelson et al., 2009; Akagi et al 2011; Alvarado et al 2010; Juncosa Calahorrano 2021).*

Reply: Please see our reply to Comment 2.

*5. L195 Section 3.4 Did you used a plume specific NO2 background (i.e., for the background condition the day the plume was retrieved by TROPOMI)? For locations with fire seasons that last for months, background conditions can change because of the presence of dilute smoke in the area.*

Reply: The background $NO_2$ is fire specific. We define background $NO_2$ as mean TROPOMI $NO_2$ columns 3 to 30 days before and after the fire. For locations with long fire season, the background $NO_2$ columns should be larger than non-fire season. We have clarified this definition of background as follows:

We then select fires where TROPOMI $NO_2$ tropospheric columns on the fire day are at least one standard deviation higher than the mean TROPOMI $NO_2$ columns 30 days before and after the fire day (excluding the nearest four days as fires may last for several days, defined as $\Omega_{NO2\_B}$).

We only use this threshold to select candidate fires, which will not influence on the calculation of emissions and lifetimes. For calculation of emissions and lifetime of fire plume, we fit the line density as Eq. (7), and parameter B here represents the background $NO_2$ line density, which is determined by $NO_2$ value over upwind regions.

*6. L220. Please include more details on figure S1 e.g., rotation angle. Might be good to have a final figure after rotation as well.*

Reply: We have added rotation angle to Figure S1 (now Figure S3). We'd like clarify the figure is indeed the final figure after rotation. We showed an example fire plume that does **not** meet our selection criteria, because the apparent direction does not align with the wind direction (*x* axis, i.e. rotation bias > 30˚). We have revised Figure S1 to show two plumes: one with small bias that satisfies the criterium, and the other with large bias that is not selected as candidate fire:

[Figure]

Figure S3 Illustration of two fire plumes with the absolute rotation biases (a) less and (b) greater than 30˚. We define the rotation biases as the angle of the two red lines. The right plume is not selected because it does not align with the wind direction (i.e., rotation bias = -44˚)

*7. L221 Can you please explain how you differentiate between the fire center vs. the apparent fire center? Is the first based on the FRP and the latter based on visual inspection? Also, this line needs more detail. What do you mean by "give good fitting statistics"? What is the correlation that needs to satisfy R2>0.5?*

Reply: Fire center is defined as where $x = 0$, and the apparent fire center is obtained from EMG function as $\mu_x$. The first is based on FRP, and the latter is based on EMG function (Equation 7). Good fitting statistics refer to the three criteria listed after that: 1) $R^2 > 0.5$; 2) $\sigma_x < x_0$; 3) $|\mu_x| < 50$ km. $R^2$ is the Pearson correlation coefficient between fitted and the observed $NO_2$ line density.

*8. Figure 1. I would have expected more tropical fires detected by TROPOMI. Did one of the criteria to remove plumes excluded those? Why? Also, it looks that there are not many fires in*

*the equatorial line in South America and Africa, which is odd. Can you explain please? Perhaps this is an issue with the satellite retrievals?*

Reply: We have implemented strict criteria to select candidate fires suitable for EMG approach. The EMG approach works best for isolated point source with clear plume patterns. In tropical regions, fires often expand over a wide region, which is better considered as area source rather than point source. Satellite retrieval may also be affected by cloud and smoke. We only select satellite retrievals with good quality and low cloud fraction (Sect. 3.4).

*9. L235 How did you identify Ag fires?*

Reply: We classify the fire episodes based on the fuel classification in the Global Fire Emission Database (GFED), which is estimated using the MODIS land cover type product. Agricultural fires refer to fires occur over cropland.

*10. L247 and Figure 1 Very nice section. I was just wondering why you didn't remove the small fire towards the upper left side of the bigger fire. I assumed your criteria will remove it because it is clearly overlapping with the bigger one. Please explain.*

Reply: Our filtering algorithm can filter out fire plumes **not** overlapping with the center plume. The small fire shown in the figure overlaps with the center plume, so they are considered as a single plume. A supplementary figure is added to help explain the grouping procedure:

[Figure]

*Figure S2 Illustration of the processes that identify and filter out nearby plumes.*

*11. L267 This sentence is confusing. I would remove the first part (before the comma) from this sentence.*

Reply: We have revised the sentence as follows:

MODIS FRP, which represents the radiant energy released by fires, has been used to approximate the biomass burned consumption in top-down emission inventories such as Global Fire Assimilation System (GFAS; Kaiser et al., 2012). We define the emission coefficient (EC) as the mass of pollutant emitted per unit of radiative energy (i.e., Emissions/FRP), which has been used to derive the emissions of chemical species from fires (Ichoku and Kaufman, 2005; Mebust et al., 2011; Mebust and Cohen, 2014).

*12. L284 I am a little concerned about the conclusion that emissions and Fire Radiative Power correlate. In past field campaign, its being shown that reduced vs. oxidized emissions of nitrogen correlate very well with the Modify Combustion Efficiency (MCE) (i.e., smoldering vs. flaming) but it has been difficult to correlate them directly to FRP. The fire condition can also impact the chemistry in PECANS. If the fire has lower MCEs, there will be a lot of reduced nitrogen (e.g., NH3) compared to oxidized nitrogen (e.g., HONO that produces OH, NOx, etc). I know that getting MCE from many different fires is challenging, but I am not convinced that using FRP is the right approach. At the very least, there should be more discussion about how MCE affects emissions in the manuscript.*

Reply: We did not force correlation between FRP and $NO_x$ emissions. Our data show reasonably good correlation between FRP and satellite-derived $NO_x$ emissions for a large ensemble of fires, suggesting variation in fire $NO_x$ emissions is largely driven by variation in FRP rather than other factors such as MCE. FRP has been used as a 'top-down' approach in biomass burning emission inventories (e.g. GFAS, FEER), which assumes an empirical linear relationship between emission rate and the amount of heat released by fires (FRP), and it does not need information on the MCE (Ichoku and Ellison, 2014). The validity of the relationship has been verified in laboratory (Wooster et al., 2005; Freeborn et al., 2008), field experiments (Wiggins et al., 2020) and satellite observations (Ichoku and Kaufman, 2005). We have added the following discussions on FRP:

Here we assume linear relationship between emission rate and FRP. While the validity of the relationship has been verified in laboratory (Wooster et al., 2005; Freeborn et al., 2008), field experiments (Wiggins et al., 2020) and satellite observations (Ichoku and Kaufman, 2005), the choice of the mass-to-energy conversion factor ($K_r$) slightly differ in Wooster et al. (2005, 0.368 g/MJ) and in Freeborn et al. (2008, 0.453g/MJ), suggesting an uncertainty of order 10% for the value of $K_r$.

*13. L333 Here is where I realized that the lifetime included dilution. Please include a few sentences somewhere earlier in the manuscript identifying all the loss processes that the estimated NO2 lifetime includes.*

Reply: The calculated lifetime should be representative of chemical lifetime if the transport speed is uniform, the direction is constant and deposition is negligible (De Foy et al., 2014). We have revised the sentence as follows:

Here we use the EMG approach to derive an effective $NO_x$ lifetime of the entire fire plume. Chemical nonlinearities can result in an effective chemical lifetime that is averaged over the plume where at each point in the plume evolution a different chemical lifetime occurs. Besides, the effective lifetime in practice will be confounded by the mixing such as those plume movement in different directions that reduces the line density.

*14. L465 I think you should discuss the thermal dependency of PAN. If the plume is injected high enough, PAN can be stable and thus its transportation can be very efficient (e.g., not a source of NOx close to the plume, at least in the time scales this relevant to this manuscript).*

Reply: That's a good point. We've added the following discussions:

In the lower troposphere, PAN is generally considered as a sink of $NO_x$ near the fire, but a source of $NO_x$ over downwind region, which deepens the gradient of $NO_2$ near the source but flattens the gradient downwind (Valin et al., 2013). For those large biomass burning events that inject PAN into upper troposphere, PAN acts as a stable reservoir of $NO_x$, leading to long-range transport of $NO_x$ (Tereszchuk et al., 2013).

*15. L468 We know well that PAN forms rapidly in fire smoke plumes and that its production plateaus after ~4 after of plume aging (Yokelson et al., 2009, Akagi et al., 2011, Alvarado et al., 2010, Juncosa Calahorrao 2021). It might be helpful to look at how the ratio of PAN/NOx changes as the plume ages and have some discussion about it.*

Reply: Thanks for the suggestion. We have added some discussions on PAN formation:

In the presence of PAN formation, we find the EMG approach tends to overestimate $NO_x$ lifetime at low $NO_x$ emission (< 5000 g/s), in which the flattening effect of PAN is more evident, while underestimating $NO_x$ lifetime at high $NO_x$ emissions and low $P(HO_x)$, in which the deepening effect takes over.
In-situ measurements show rapid formation of PAN in young smoke within 4 hours of aging, and PAN contributes about 25% of the total reactive nitrogen (Alvarado et al., 2010; Juncosa Calahorrano et al., 2021), suggesting a non-negligible sink of PAN for $NO_x$ near fire source.

*16. L470 can you find another word for "estimate" so it is not right before "overestimate"*

Reply: We have revised the sentence as follows:

Overall, we assess that the overestimate at low $NO_x$ emissions (< 5000 g/s) cause around 33% negative biases to the derived emissions, while 18% positive biases at high $NO_x$ emissions (> 5000 g/s).

**References:**

Akagi, S. K., Craven, J. S., Taylor, J. W., McMeeking, G. R., Yokelson, R. J., Burling, I. R., Urbanski, S. P., Wold, C. E., Seinfeld, J. H., Coe, H., Alvarado, M. J., and Weise, D. R.: Evolution of trace gases and particles emitted by a chaparral fire in California, Atmos Chem Phys, 12, 1397–1421, https://doi.org/10.5194/acp-12-1397-2012, 2012.

De Foy, B., Wilkins, J. L., Lu, Z., Streets, D. G., and Duncan, B. N.: Model evaluation of methods for estimating surface emissions and chemical lifetimes from satellite data, Atmos Environ, 98, 66–77, https://doi.org/10.1016/j.atmosenv.2014.08.051, 2014.

Freeborn, P. H., Wooster, M. J., Hao, W. M., Ryan, C. A., Nordgren, B. L., Baker, S. P., and Ichoku, C.: Relationships between energy release, fuel mass loss, and trace gas and aerosol

emissions during laboratory biomass fires, J Geophys Res Atmospheres 1984 2012, 113, https://doi.org/10.1029/2007jd008679, 2008.

Juncosa Clahorrano, J., Lindaas, J., O'Dell, K., Palm, B. B., Peng, Q., Flocke, F., Pollack, I. B., Garofalo, L. A., Farmer, D. K., Pierce, J. R., Collett, J. L., Weinheimer, A., Campos, T., Hornbrook, R. S., Hall, S. R., Ullmann, K., Pothier, M. A., Apel, E. C., Permar, W., Hu, L., Hills, A. J., Montzka, D., Tyndall, G., Thornton, J. A., and Fischer, E. V.: Daytime Oxidized Reactive Nitrogen Partitioning in Western U.S. Wildfire Smoke Plumes, J Geophys Res Atmospheres, 126, https://doi.org/10.1029/2020jd033484, 2021.

Ichoku, C. and Ellison, L.: Global top-down smoke-aerosol emissions estimation using satellite fire radiative power measurements, 14, 6643–6667, https://doi.org/10.5194/acp-14-6643-2014, 2014.

Ichoku, C. and Kaufman, Y. J.: A Method to Derive Smoke Emission Rates From MODIS Fire Radiative Energy Measurements, Ieee T Geosci Remote, 43, 2636–2649, https://doi.org/10.1109/tgrs.2005.857328, 2005.

Velde, I. R. van der, Werf, G. R. van der, Houweling, S., Eskes, H. J., Veefkind, J. P., Borsdorff, T., and Aben, I.: Biomass burning combustion efficiency observed from space using measurements of CO and NO2 by the TROPOspheric Monitoring Instrument (TROPOMI), Atmos Chem Phys, 21, 597–616, https://doi.org/10.5194/acp-21-597-2021, 2020.

Wiggins, E. B., Soja, A. J., Gargulinski, E., Halliday, H. S., Pierce, R. B., Schmidt, C. C., Nowak, J. B., DiGangi, J. P., Diskin, G. S., Katich, J. M., Perring, A. E., Schwarz, J. P., Anderson, B. E., Chen, G., Crosbie, E. C., Jordan, C., Robinson, C. E., Sanchez, K. J., Shingler, T. J., Shook, M., Thornhill, K. L., Winstead, E. L., Ziemba, L. D., and Moore, R. H.: High Temporal Resolution Satellite Observations of Fire Radiative Power Reveal Link Between Fire Behavior and Aerosol and Gas Emissions, Geophys Res Lett, 47, https://doi.org/10.1029/2020gl090707, 2020.

Wooster, M. J., Roberts, G., Perry, G. L. W., and Kaufman, Y. J.: Retrieval of biomass combustion rates and totals from fire radiative power observations: FRP derivation and calibration relationships between biomass consumption and fire radiative energy release, 110, 349–24, https://doi.org/10.1029/2005jd006318, 2005.

**Reply to Reviewer 2**

*This paper estimates biomass burning NOx emissions and lifetime using daily observations from TROPOMI. The topic has a broad interest, and the investigation is solid. I particularly appreciate the validation using a plume model. The work suggests decreasing NOx lifetime with fire intensity due to the increase in both NOx abundance and hydroxyl radical production. I would recommend minor revision before publication.*

Reply: We would like to thank the reviewer for their constructive feedback and time spent reviewing this paper. Below is our response to reviewer's comments.

***General comments.***
*1. Section 3.1 profile correction using GEOS-CF: How sensitivity the derived lifetime and emissions to this correction?*

Reply: We have shown in Sect. 4.3 that using GEOS-CF profiles largely enhances the $NO_2$ column density and the derived emissions. Figure 4 shows that using updated profile increases the derived emission factors. Figure S5 shows the relationship between emissions and FRP using original TROPOMI $NO_2$ data without profile correction. The derived $NO_x$ lifetime is not sensitive to the profile correction, largely because the lifetime is determined by the shape of the fire plumes that are not affected by the *a priori*. We have added a new supplementary figure that shows the derived $NO_x$ lifetime using the original TROPOMI data:

[Figure]

Figure S7 Same as Figure 5 but using original TROPOMI $NO_2$ data without updating the a priori profile.

We have added the following discussions on the sensitivity of derived NO$_x$ lifetime to profile correction:

We find similar NO$_x$ lifetime using original TROPOMI NO$_2$ data, largely because the derived NO$_x$ lifetime is determined by the shape of fire plumes that are not affected by the *a priori*.

*2. Section 3.2. It is not very clear to me how ALH is related to the EMG approach. I suppose the authors indicate a consistent wind layer height and injection height. If so, I would suggest making this clearer in the text. Are the derived results from the EMG approach very sensitive to the choice of wind layer heights? Additional sensitivity analysis would be beneficial to the study.?*

Reply: We'd like to clarify that we did **not** use a consistent wind layer height. ALH is used to determine the fire injection height. Assuming consistent wind layer height generally works fine for small fires, but big fires are often associated with high injection height. We clarify this in the main text:

Previous studies either use the averaged wind speed of the first several layers (Beirle et al., 2011; Lu et al., 2015) or choose a constant layer such as 900 hPa (Mebust et al., 2011), but injection height of wildfires varies significantly, especially for large fires which inject emissions into high altitudes (Val Martin et al., 2010). To account for varying injection height, we use TROPOMI ALH as an approximation of the fire injection height instead of assuming a constant layer. We vertically interpolate ERA-5 wind data to the pressure level of aerosol layer. For the fires without valid ALH (~36% of the selected fires), we use 900hPa, as the ALP level for the majority of selected fires is near 900 hPa (see Sect. 4.1).

We have conducted sensitivity analysis in Section 5.3. We estimate that an increase of 500 m ALH corresponds to ~22% increase of the wind speed on average, meaning that NO$_x$ lifetime will decrease by ~18%. In Sect. 5.3, we have discussed how EMG approach is sensitive to the choice of wind layer heights:

Uncertainty and variance of the wind speed, however, should lead to errors in the derived NO$_x$ lifetime. Here we determine the wind speed by interpolating the wind profile to TROPOMI derived aerosol layer. Comparison of TROPOMI ALH with plume height from the Cloud-Aerosol LIdar with Orthogonal Polarization (CALIOP) and the Multi-angle Imaging SpectroRadiometer (MISR) measurements suggest that TROPOMI ALH is overall 500 m lower (Griffin et al., 2020; Nanda et al., 2020). We estimate that an increase of 500 m ALH corresponds to ~22% increase of the wind speed on average, meaning that NO$_x$ lifetime will decrease by ~18%.

*3. Section 3.4. "pixels are grouped to separate plumes based on their connections with surrounding pixels" I recommend a diagram or plot here to illustrate the grouping algorithm. It is not easy for me to get it from the text here.*

Reply: We have added a figure in Supplement to illustrate the processes of the grouping algorithm:

[Figure]

Figure S2 Illustration of the processes that identify and filter out nearby plumes.

*4. Section 4.5. I'm surprised to see the results for the wind speed less than 2 m/s. As far as my understanding, the EMG function is not suitable for the cases of calm winds. Is there any special reason for applying EMG for calm-wind conditions?*

Reply: We did not apply filtering for wind speed, because the definition of 'calm' condition may be subjective. We assume EMG function should be able to identify the suitable cases as long as a satisfying fitting is achieved. There are indeed very few fire cases selected for calm winds (< 4%). To avoid confusion, we have removed the fires with wind speed less than 2m/s in Figure 5:

[Figure]

Figure 5 The mean and standard deviation of TROPOMI derived NO$_x$ lifetime from fires at different emissions (colour) and wind speeds. Fire episodes with less than 2 m/s wind speed are not shown.

**Specific comments.**

*5. Page 2, line 49. Please add reference for "the improved signal-to-noise ratio". I would*

*suggest more details about the improved signal-to-noise ratio to justify the usage of daily observation. For example, how does the ratio improve from OMI to TROPOMI? How does one TROPOMI observation compare to several OMI observations? It is not necessary to discuss this in the abstract, but somewhere in the main text would be appreciated.*

Reply: We have added references for TROPOMI:

The finer spatial resolution (~7 × 3.5 km$^2$), and the improved signal-to-noise ratio of TROPOMI compared to OMI offer new opportunities to more reliably interpret observations of individual plumes (Veefkind et al., 2012; Judd et al., 2019; van Geffen et al., 2020).

We also include a new supplementary figure that directly compares daily TROPOMI vs. OMI observation for detecting fire NO$_x$. The figure clearly shows the improved performance of TROPOMI over OMI:

[Figure]

Figure S4 Maps of TROPOMI (left) and OMI (right) tropospheric NO$_2$ over Australia on October 21, 2018. The figures are acquired from TEMIS: https://www.temis.nl/airpollution/no2col/ . The red box labels the location of the fire episode shown in Figure 2.

We have added the following discussions in the main text:

Several NO$_2$ plumes are detected by TROPOMI on this day, which outperforms OMI observation on the same day which detects less smaller fires, shows less spatial gradients and larger data gap (Figure S4).

*6. Page 6, line 180. Please add reference for PECANS. Additionally, please clarify the reasons for the settings in the model, such as the diffusion coefficients and O3 concentrations.*

Reply: We have added reference for PECANS:
To understand the factors that control the NO$_x$ lifetime, we employ a one-dimensional (1-D) multi-box plume model based on the Python Editable Chemical Atmospheric Numerical Solver (PECANS; Laughner and Cohen, 2019; Laughner 2019).

We have clarified the reasons for settings:

The wind speed is fixed at 5 m/s, and the diffusion coefficients are also fixed at 100 $m^2$/s following Laughner and Cohen (2019).

The $O_3$ concentration is fixed at 65 ppbv, which is close to observed mean $O_3$ concentration near fire plumes (Alvarado and Prinn, 2009; Alvarado et al., 2014). A fixed branching ratio to form $RONO_2$ in $RO_2 + NO$ reaction of 0.05 is used following Laughner and Cohen (2019).

*7. Line 227. Please clarify the details of the 50 initial conditions.*

Reply: We have revised the sentence as follows:

To test the sensitivity of the fitting results to initial conditions, we repeat the fitting with varying initial values for each parameter 50 times, and we exclude fires where the standard deviation of resulting emissions is more than 50% of the emissions.

*8. Line 305. Does Mebust and Cohen (2014) adopt a similar method as this study? If not, I would suggest rephrasing this part by mentioning the results using standard TROPOMI products firstly and then comparing with that of Mebust and Cohen (2014). Otherwise, the readers may get confused here.*

Reply: Mebust and Cohen (2014) use different method as this study. We have revised this part to avoid confusion:

Using the standard TROPOMI $NO_2$ products without updating the *a priori* profile, the derived $NO_x$ EFs are 44 to 66% of $EF_{sat}$, and 26 to 68% of $EF_{andreae}$. Assessment of TROPOMI $NO_2$ with *in situ* measurements also suggest TROPOMI $NO_2$ is biased low over polluted regions, and replacing the coarse-resolution *a priori* profile with fine-resolution simulations could largely reduce the low biases (Judd et al., 2020; Tack et al., 2021). Our derived $NO_x$ EFs are nearly 3 times larger than a previous study based on OMI observations, which suggest $NO_x$ EFs are lower than 1g/kg in all fuel types (Mebust and Cohen, 2014). Besides the differences in satellite instruments and methods, the discrepancy is partially due to less accurate representation of biomass burning emissions in the *a priori* profile of $NO_2$ in Mebust and Cohen (2014). Using the standard TROPOMI $NO_2$ products without updating the *a priori* profile, the derived $NO_x$ EFs are similar to those developed by Mebust and Cohen (2014) for boreal and temperate forest fires, but still higher over other fuel types.

*8. Section 5.3. please clarify the calculation of chemical lifetime.*

Reply: We have revised the sentence as follows:

To assess if EMG fitted $NO_x$ lifetime is indicative of the chemical lifetime, we use the PECANS model to calculate an EMG fitted lifetime and a chemical lifetime of $NO_x$ from

two permanent losses of $NO_x$ through the formation of $HNO_3$ and $RONO_2$ over downwind region (*i.e.*, mean $NO_x$ concentration divided by the mean chemical loss of $NO_x$).

**Reply to Reviewer 3**

*The authors directly estimate NOX emissions and lifetime for fires by using an exponentially modified Gaussian analysis of tropospheric NO2 columns observed by TROPOMI. The authors firstly correct the low bias of TROPOMI retrieved NO2 columns by replacing the a priori profile of NO2 with the GEOS-CF simulated profile at a finer resolution of 0.25. Representative NOx emission factors for six fuel types are derived by using the observations of fire radiative power from MODIS. The authors also discussed the uncertainties and capabilities of the method thoroughly. The scope fits ACP and the scientific idea is new. I recommend the paper be published after the authors address the following comments.*

> Reply: We would like to thank the reviewer for their constructive feedback and time spent reviewing this paper. Below is our response to reviewer's comments.

***Major comments.***
*1. A better result is expected after the authors used a priori NO2 profile with 0.25 to replace the one used by the operational product. However, this spatial resolution is still much coarser than TROPOMI's, which implies that nearby pixels use the same profile shape. As the authors presented in the paper that fire events normally take place locally. How much uncertainty can it contribute?*

> Reply: We group fire pixels whose distances are within 20 km as a single fire event (~ 10 km radius). For most fire plumes we analyzed, the extent of the fire plumes is larger than the resolution of GEOS-CF (0.25˚). We agree that such resolution may not be able to resolve the spatial variation of $NO_x$ within fire plumes, but the focus here is to derive an overall emission and lifetime estimate for the entire plume. As a result, we expect the *a priori* is largely converged with respect to spatial resolution.

> We have added the following discussions in the revised manuscript:

> The resolution of current global model simulation, however, is not sufficient to resolve the fine-scale chemical evolution of fire plumes, and better treatment of the fire injection is needed (Paugam et al., 2016). Assessment of the satellite retrieval uncertainty will benefit from high-resolution regional simulations combined with *in situ* measurements that sample individual fire smokes from the point of emission to downwind regions (Juncosa Clahorrano et al., 2021; Lindaas et al., 2020).

*2. This plume-based method works only when the wind speed is not small, that is the plume exists. The authors keep every case even with very low wind speeds (< 2m/s). Should these cases be removed for the scientific reason?*

> Reply: We did not apply filtering for wind speed because the definition of 'calm' condition may be subjective. We assume EMG function should be able to identify the suitable cases as long as a satisfying fitting is achieved. There are indeed very few fire cases selected for calm

winds (< 5%). To avoid confusion, we have removed the fires with wind speed less than 2m/s in Figure 5:

[Figure]

Figure 5 The mean and standard deviation of TROPOMI derived $NO_x$ lifetime from fires at different emissions (colour) and wind speeds. Fire episodes with less than 2 m/s wind speed are not shown.

*3. The authors argue that the difference between TROPOMI and OMI derived is mainly due to the a priori $NO_2$ profiles, which is not accurate. The VCDs of TROPOMI are found to be lower than OMI's over many places, which is mainly caused by impropriate surface albedos or cloud pressures. Please give more solid proof if the authors want to draw this conclusion (i.e. section 4.4).*

Reply: We agree that the *a priori* is not the only reason for the difference between TROPOMI and OMI. As we did not use OMI observations in this manuscript, the direct comparison between TROPOMI and OMI retrievals is beyond the scope of this study. We have revised the section to avoid confusion:

Using the standard TROPOMI $NO_2$ products without updating the *a priori* profile, the derived $NO_x$ EFs are 44 to 66% of $EF_{sat}$, and 26 to 68% of $EF_{andreae}$. Assessment of TROPOMI $NO_2$ with *in situ* measurements also suggest TROPOMI $NO_2$ is biased low over polluted regions, and replacing the coarse-resolution *a priori* profile with fine-resolution simulations could largely reduce the low biases (Judd et al., 2020; Tack et al., 2021). Our derived $NO_x$ EFs are nearly 3 times larger than a previous study based on OMI observations, which suggest $NO_x$ EFs are lower than 1g/kg in all fuel types (Mebust and Cohen, 2014). Besides the differences in satellite instruments and methods, the discrepancy is partially due to less accurate representation of biomass burning emissions in the *a priori* profile of $NO_2$ in Mebust and Cohen (2014). Using the standard TROPOMI $NO_2$ products without updating the *a priori* profile, the derived $NO_x$ EFs are similar to those developed by Mebust and Cohen (2014) for boreal and temperate forest fires, but still higher over other fuel types.

*4. The section 3.4 is long and complicated. A flowchart is helpful to explain the procedure or moving this part to the supplementary.*

Reply: We have added a flowchart in the Supplement to explain the procedure of fire selection:

[Figure]

Figure S1 Flowchart that illustrates the processes to select candidate fires.

*5. The authours intend to derive representative NOX emission factors for six fuel types. However, satellite observations are available once per day, and some fire events can last for several days. In these cases, the fire intensity and the chemical condition also change. The authours, at least, should give an example to explain how to consider the emission factor for a certain fuel type.*

Reply: The fuel type classification is based on annual MODIS land cover type product, which is a function of location only, and should not vary temporally. We have clarified this point in the revised manuscript:

The fire episodes are classified based on MODIS detected fire location following the fuel classification in the Global Fire Emission Database (GFED), which is estimated using the MODIS land cover type product and University of Maryland classification scheme (Friedl et al., 2010; van der Werf et al., 2017).

We agree that fire intensity and the chemical condition may change, which cannot be captured by the single overpass of TROPOMI. We derive $NO_x$ emission factor by matching TROPOMI derived $NO_x$ emissions with concurrent MODIS FRP, which is considered as a good indicator of fire intensity (Ichoku and Kaufman, 2005; Wiggins et al., 2020). Since the lifetime of $NO_x$ is within several hours (Figure 5), the influence of fire emissions from previous days should be negligible. We have clarified the limitation of TROPOMI as follows:

TROPOMI is limited to single overpass per day, which cannot resolve the short-term evolution of fire plumes. The newly launched or upcoming geostationary satellite instruments such as GEMS and TEMPO will offer an unprecedented opportunity to continuously observe the emissions and chemical evolution of $NO_x$ from fires that will no longer be limited to a single snapshot (Chance et al., 2013; Kim et al., 2019).

**Specific comments.**

*1. Line 11: "behaviour" should be "behavior". "occur" should be "that occur".*

Reply: Done.

*2. Line 15: The sentence is a little confusing. I think the authors recalculate the NO2 VCD of every pixel with the GEOS-CF simulated profile not only over the fire plumes?*

Reply: It's true that we apply the GEOS-CF profile for every pixel, but overall such replacement only increases the $NO_2$ VCD for plume affected pixels. We have revised the sentence as follows:

We update the *a priori* profile of $NO_2$ with a fine-resolution (0.25˚) global model simulation from NASA's GEOS Composition Forecasting System (GEOS-CF), which largely enhances $NO_2$ columns over fire plumes.

*3. Line 24 and 27: Please list enough examples when you give examples.*

Reply: We have added more examples as suggested:

Biomass burning emissions affect global radiative forcing, the hydrological cycle, ecosystem and air quality (e.g., Crutzen and Andreae, 1990; Penner et al., 1992; Johnston et al., 2012; Liu et al., 2014).

Biomass burning emissions inventories used in models are subject to uncertainties in estimates or measurements of the burned area, fuel loadings, combustion efficiency, and also the compound-specific emission factors that relate the mass of a chemical species emitted to fuel consumption (e.g., Petrenko et al., 2012; Liu et al., 2020; Carter et al., 2020).

*4. Line 42: "the fire detection".*

Reply: Done.

*5. Line 44: "has a finer spatial resolution".*

Reply: Done.

*6. Line 46: "spatial resolutions"*

Reply: Done.

*7. Line 57 and 64: Please cite recent and more relevant studies about TROPOMI.*

Reply: We have added more recent studies about TROPOMI:

The accuracy of satellite retrieval of $NO_2$ columns largely depends on the *a priori* knowledge of $NO_2$ vertical profile shape needed for calculating air mass factor (e.g., Boersma et al., 2018; Verhoelst et al., 2021).

Replacing the *a priori* vertical profile from a fine-resolution regional model can enhance the spatial gradient and correct the low bias of satellite retrieved $NO_2$ over polluted regions (e.g., Russell et al., 2011; Valin et al., 2011; Goldberg et al., 2017; Ialongo et al., 2020; Judd et al., 2020; Tack et al., 2021).

*8. Line 83: "afternoon global" is quite obscure, please specify the overpassing time is around 13:30 local time.*

Reply: Done. Revised as follows:

TROPOMI provides afternoon (~ 1:30 PM local time) global observations in the UV−visible−near infrared−shortwave spectra with a fine spatial resolution of $7 \times 3.5$ km$^2$ at nadir (increased to $5.5 \times 3.5$ km$^2$ since August 2019).

*9. Line 86-93: Do you use the S5P operational product that retrieved by KNMI? If so, you should cite van Geffen et al., (2019) when introducing the way of retrieval. Besides, you should also cite the validation paper (i.e. Tijl et al., 2021 https://amt.copernicus.org/articles/14/481/2021/) when discussing the underestimation of S5P.*

*van Geffen, J. H. G. M., Eskes, H. J., Boersma, K. F., Maasakkers, J. D., and Veefkind, J. P.: TROPOMI ATBD of the total and tropospheric NO2 data products (issue 1.4.0), Royal Netherlands Meteorological Institute (KNMI), De Bilt, the Netherlands, 2019.*

Reply: We have added the suggested references.

*10. Line 165: The format of the reference is "Laughner and Cohen, (2019)".*

Reply: Done.

*11. Line 203: Not very clear to me why "3 to 30 days before and after the fire day" is 56 days in total.*

Reply: We include the days 30 days before and after the fire day, which is 60 days, and we further exclude two days before and after the fire because fire often lasts for several days, which gives 56 (i.e., 60 – 4 = 56) days in total. We have revised the sentence as follows:

We then select fires where TROPOMI $NO_2$ tropospheric columns on the fire day are at least one standard deviation higher than the mean TROPOMI $NO_2$ columns 30 days before and after the fire day (excluding the nearest four days as fires may last for several days, defined as $\Omega_{NO2\_B}$).

*12. Line 206: "filter" should be "filters".*

Reply: Done.

*13. Line 224-225: Please give specific examples to explain "We also exclude fires in which TROPOMI NO2 line densities are monotonically increasing or decreasing within the region."*

Reply: We have revised the sentence as follows:

We only include fires in which TROPOMI $NO_2$ line densities peak near the fire centre, meaning that fires with monotonically increasing or decreasing line densities within the region are excluded.

*14. Line 241: "10000 MW" should be "10,000 MW"*

Reply: Done.

*15. The resolution of Figure 2 is too low. It's better to start with the original TROPOMI NO2 data before (a).*

Reply: We provide high-resolution figures for the revised manuscript. The resolution or the re-gridded data is 0.05˚ for the maps shown in Figure 2, which is close to the original resolution of TROPOMI ($5.5 \times 3.5$ km$^2$). We did not start with the original TROPOMI NO2 data because the first step of processing TROPOMI data is to rotate swaths data along the wind direction. The original TROPOMI data can be found from TEMIS website (https://www.temis.nl/airpollution/no2col/). We have added a supplementary figure that compares TROPOMI and OMI data for the fire episode:

[Figure]

Figure S3 Maps of TROPOMI (left) and OMI (right) tropospheric NO$_2$ over Australia on October 21, 2018. The figures are acquired from TEMIS: https://www.temis.nl/airpollution/no2.php. The red box labels the location of the fire episode shown in Figure 2.

**References:**

Ichoku, C. and Kaufman, Y. J.: A Method to Derive Smoke Emission Rates From MODIS Fire Radiative Energy Measurements, Ieee T Geosci Remote, 43, 2636–2649, https://doi.org/10.1109/tgrs.2005.857328, 2005.

Wiggins, E. B., Soja, A. J., Gargulinski, E., Halliday, H. S., Pierce, R. B., Schmidt, C. C., Nowak, J. B., DiGangi, J. P., Diskin, G. S., Katich, J. M., Perring, A. E., Schwarz, J. P., Anderson, B. E., Chen, G., Crosbie, E. C., Jordan, C., Robinson, C. E., Sanchez, K. J., Shingler, T. J., Shook, M., Thornhill, K. L., Winstead, E. L., Ziemba, L. D., and Moore, R. H.: High Temporal Resolution Satellite Observations of Fire Radiative Power Reveal Link Between Fire Behavior and Aerosol and Gas Emissions, Geophys Res Lett, 47, https://doi.org/10.1029/2020gl090707, 2020.